# Bayesian and Non-Bayesian Analysis of Exponentiated Exponential Stress–Strength Model Based on Generalized Progressive Hybrid Censoring Process

**Manal M. Yousef** [1,†] , **Amal S. Hassan** [2,†], **Huda M. Alshanbari** [3,*,†] , **Abdal-Aziz H. El-Bagoury** [4,†]
and **Ehab M. Almetwally** [5,6,†]

1   Department of Mathematics, Faculty of Science, New Valley University, El-Khargah 72511, Egypt
2   Faculty of Graduate Studies for Statistical Research, Cairo University, Giza 12613, Egypt
3   Department of Mathematical Sciences, College of Science, Princess Nourah bint Abdulrahman University, P.O. Box 84428, Riyadh 11671, Saudi Arabia
4   Basic Science Department, Higher Institute of Engineering and Technology, El-Mahala El-Kobra 6734723, Egypt
5   Faculty of Business Administration, Delta University of Science and Technology, Gamasa 11152, Egypt
6   The Scientific Association for Studies and Applied Research, Al Manzalah 35646, Egypt
*   Correspondence: hmalshanbari@pnu.edu.sa
†   These authors contributed equally to this work.

**Abstract:** In many real-life scenarios, systems frequently perform badly in difficult operating situations. The multiple failures that take place when systems reach their lower, higher, or extreme functioning states typically receive little attention from researchers. This study uses generalized progressive hybrid censoring to discuss the inference of $R = P(X < Y < Z)$ for a component when it is exposed to two stresses, $Y, Z$, and it has one strength $X$ that is regarded. We assume that both the stresses and strength variables follow an exponentiated exponential distribution with a common scale parameter. We obtain $R$'s maximum likelihood estimator and approximate confidence intervals. In addition, the Bayesian estimators for symmetric, such as squared error, and asymmetric loss functions, such as linear exponential, are developed. Credible intervals with the highest posterior densities are established. Monte Carlo simulations are used to evaluate and compare the effectiveness of the many proposed estimators. The process is then precisely described using an analysis of real data.

**Keywords:** stress–strength model; exponentiated exponential; generalized progressive hybrid censoring; maximum likelihood method; Bayesian inference

**MSC:** 62F10; 62F15; 62F03; 62N01; 62N05

## 1. Introduction

The stress–strength system is one of the most widely used data analysis methods in a variety of fields, including industrial engineering, military applications, health, and applied sciences. The stress–strength system reliability is the evaluation of a component's reliability in terms of the random variable $X$ that represents the stress of the component that is exposed to $Y$ that represents the component's strength available to overcome the possible stress. When the stress exceeds the strength of the system, it fails. Due to its practical applications, Bhattacharyya and Johnson [1] were the first researchers to be interested in investigating and deriving the reliability of the stress–strength model. In recent years, a lot of works have been conducted on the problem of estimating the stress–strength model; see, for example, Ahmad et al. [2], Saraçoglu et al. [3], Hassan et al. [4], Kotb and Raqab [5], and Jana and Bera [6].

This paper is interested in the model $R = P(X < Y < Z)$ that discusses the case where the strength $Y$ should not only be greater than stress $X$ but also be smaller than stress $Z$.

The model $R = P(X < Y < Z)$ is necessary when devices cease to function under extreme lower as well as extreme upper stress operating environments. The main idea of this model was first introduced by Chandra and Owen [7]. For example, a person's blood pressure has two limits, systolic and diastolic, and their blood pressure should lie within these limits. As another example, some electrical components malfunction when positioned below and above a specific power generator. In a similar vein, many gadgets are non-functional in both high and low temperatures. The $R = P(X < Y < Z)$ stress–strength models have a wide range of applications in engineering, psychology, genetics, clinical trials, and other fields; see Kotz et al. [8].

In the literature, Chandra and Owen [7] developed maximum likelihood estimators (MLEs) and uniform minimum unbiased estimators (UMVUEs) for $R = P(X < Y < Z)$. The lowest variance unbiased and the MLEs of $R = P(X < Y < Z)$ were investigated by Singh [9], where $X, Y,$ and $Z$ are mutually independent random variables that follow the normal distribution. Dutta and Sriwastav [10] addressed the estimation of $R$ when $X, Y,$ and $Z$ are exponentially distributed. Ahmad et al. [11] discussed the comparative inference on reliability estimation for a multi-component stress–strength model under power Lomax distribution. Almetwally et al. [12] developed an optimal censoring plan for multi-stress—strength reliability based on progressive first failure utilising Bayesian and non-Bayesian methods. Ivshin [13] presented the MLE and UMVUE of $R$ when $X, Y,$ and $Z$ are either uniform or exponential random variables. The estimation of $R = P(X < Y < Z)$ for the Weibull distribution in the presence of $k$ outliers have been provided by Hassan et al. [14]. The estimation of the stress–strength reliability for a new exponential inverted Topp–Leone distribution has been provided by Metwally et al. [15]. Estimation by using various estimation methods of the stress–strength reliability of the exponentiated inverted Weibull distribution has been obtained by Abu El Azm et al. [16]. The inference of a fuzzy stress–strength reliability model for the inverse Rayleigh distribution has been discussed by Sabry et al. [17].

Under the assumption that the three samples were independent, Wang et al. [18] used nonparametric normal approximations and the Jackknife empirical likelihood to make a statistical inference for $R$. Patowary et al. [19] used the Monte Carlo simulation to study the technique of the reliability estimation for $R = P(X < Y < Z)$ of $n$ standby systems ($n = 1, 2$). Yousif et al. [20] proposed the estimation reliability for $R = P(X < Y < Z)$ using the exponentiated inverse Rayleigh distribution. Hameed et al. [21] discussed the estimation of $R = P(Y_1 < X < Y_2)$ using the inverse Kumaraswamy distribution. Attia and Karam [22] studied the Bayesian estimation of $R = P(X < Y < Z)$ for the Dagum distribution. Some estimation methods of $R = P(X < Y < Z)$ for the inverse Rayleigh distribution were considered by Raheem et al. [23] and Abd Elfattah and Taha [24]. Yousef and Almetwally [25] provided different estimates for a multi-stress–strength model when data are observed from the Kumaraswamy distribution.

Life testing studies, which can be characterized as mathematical and statistical models of survival analysis, are frequently employed in engineering, biology, mechanical, and other disciplines of research. In fact, we are unable to observe the failure time of all the units due to a variety of constraints, such as time and cost. Before all the observations fail, it is typical to stop in the middle of the process. Type-I and Type-II censoring schemes are the two most common censoring schemes among all censoring cases (see, for example, Meeker and Escobar [26]).

Before the final termination, an inevitable pause or loss of the experiment units is likely to occur. The constraint in those two censoring techniques, however, is that the units cannot be removed during the trial. Cohen [27] initially proposed a progressive censoring strategy to address this inflexibility. According to the progressive Type-II censored schemes defect, if the experimental units are highly trustworthy, this experiment will last a long period. As a result, Kundu and Joarder [28] proposed the progressively hybrid censoring scheme. Moreover, El-Sherpieny et al. [29] introduced progressive Type-I and Type-II hybrid censored schemes based on the maximum product spacing method as an alterna-

tive estimation method for this scheme. The implementation of $n$ independent identical distributed units is employed for the censoring scheme. At $\min(T, X_m)$, the experimenter will terminate the operation. The time $T$ as well as $1 \leq m \leq n$ are predetermined in this case. In the context of the progressive type-II censored approach, the experiment span will not be more than $T$. However, the observations we acquired would be insufficient if the predetermined termination time $T$ is small. Cho et al. [30] presented a new censoring scheme called the generalized progressive hybrid censoring (GPHC) scheme, which allows us to obtain a specified series of failures.

The estimation of the parameters for the Gompertz distribution was obtained using the ML and the Bayesian methods under different loss functions based on Mohie El-Din et al. [31]. Tu and Gui [32] considered the estimation for the Kumaraswamy distribution under the GPHC. Nagy et al. [33] used a GPHC sample from the Burr XII distribution to estimate the unknown parameters, reliability, and hazard functions. Maswadah [34] improved the ML estimation method using the Runge–Kutta technique.

The goal of this paper is to establish the reliability inferences for the stress–strength variables that follow the exponentiated exponential distributions (EEDs) with a similar scale parameter. As far as we know, no earlier work has attempted to estimate $R$ using an EED under the GPHC scheme. As a result, the estimations of $R$'s statistical inference based on data from a GPHC scheme will be discussed in this study. The lifetime distributions of the one strength $Y$ and two stresses, $X$ and $Z$, are considered to have independent EEDs with the same scale parameter. MLEs as well as Bayesian estimators under different loss functions of $R$ are derived. The asymptotic confidence intervals as well as Bayesian credible intervals are created. To compare the different methodologies, extensive simulations were run, and dataset was evaluated for demonstration reasons.

This paper proceeds as follows: First, data description and reliability model are mentioned in Section 2. The MLEs as well as the approximate confidence intervals of $R$ will be derived in Section 3. We calculate the Bayesian estimators with different loss functions in Section 4. Furthermore, we apply the Markov chain Monte Carlo (MCMC) algorithm to derive the Bayesian estimators and establish the highest posterior density (HPD) intervals under the sample generated by the MCMC algorithm in Section 5. Then, in Section 6, the data analysis is demonstrated. Finally, conclusions are arranged in the last Section 7.

## 2. Data Description and Reliability Model

Ahuja and Nash [35] presented the two-parameter EED which was subsequently investigated by Gupta and Kundu [36]. The shape and scale parameters of this distribution are similar to those of the gamma and Weibull distributions. In many circumstances, it has a better fit than the Weibull and Gamma distributions (Raja and Mir [37]). The applications of the EED were substantial, and we point out: models to determine bout standards for analysis of animal conduct (Yeates et al. [38]); software reliability increase fashions for vital best metrics (Subburaj et al. [39]); models for episode height and period for eco-hydro-climatic packages (Biondi et al. [40]); and therapy price modeling (Kannan et al. [41]). The probability density function (PDF) and cumulative distribution function (CDF) of EED are defined as

$$f(x) = \alpha \lambda e^{-\lambda x}(1 - e^{-\lambda x})^{\alpha - 1}; \qquad x > 0, \quad \alpha, \lambda > 0, \tag{1}$$

and

$$F(x) = (1 - e^{-\lambda x})^{\alpha}, \tag{2}$$

respectively, for $x > 0, \alpha > 0$ and $\lambda > 0$. We write $X \sim \mathrm{EED}(\lambda, \alpha)$, and here, $\alpha$ is the shape parameter and $\lambda$ is the scale parameter.

Let $X \sim \text{EED}(\lambda, \alpha_1)$, $Y \sim \text{EED}(\lambda, \alpha_2)$, and $Z \sim \text{EED}(\lambda, \alpha_3)$, and they are independent. The reliability formula of the stress–strength model that the probability of a component strength falling in between two stresses is given by:

$$R = \int_0^\infty \int_x^\infty \int_x^z f(x)f(y)f(z)dydzdx,$$
$$= \frac{\alpha_2\alpha_3}{(\alpha_1 + \alpha_2)(\alpha_1 + \alpha_2 + \alpha_3)}. \tag{3}$$

The reliability $R$ of stress–strength model with different values of parameters, see Figure 1.

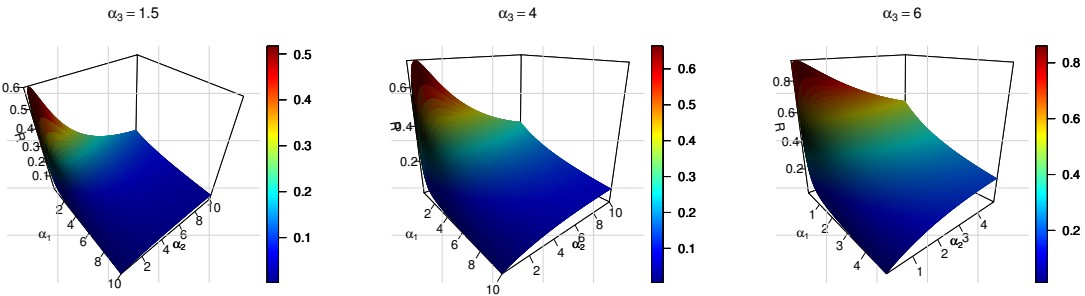

**Figure 1.** The reliability $R$ of stress–strength model with different values of parameters.

Figure 1 shows the reliability $R$ of the stress–strength model where the higher the value of parameter $\alpha_3$, the higher the reliability value. We also note that with the change in the parameter values, the reliability rating changes, and in most matters, it gives high results and covers most of the values.

Assume that our research group is made of $n$ independent units with the same lifetime distribution, where $X_1, X_2, \ldots, X_n$ represent the corresponding lifetime. The integers $k$ and $m$, $k < m$, as well as $R_1, R_2, \ldots, R_m$, which can satisfy the equation $\sum_{i=1}^m R_i + m = n$ function as preplanned integers, have been under predetermination between zero and $n$. On the arrival of the first failure $X_1$, we withdraw $R_1$ units. When the second failure, $X_2$, occurs, we remove $R_2$ units at random from the $n - 2 - R_1$ survivors. With the rest of the survival units removed, the procedure is repeated and ended at $T^* = \max(\min(T, X_m), X_k)$. It vastly improved prior approaches by allowing us to choose whether or not to continue the experiment if the sample size is insufficient at the predetermined cut-off time $T$. Researchers would prefer to obtain $m$ failures under the GPHC scheme, but they can alternatively choose $k$ failures, which are considered the bare minimum. The GPHC scheme is referred to as $R_1, R_2, \ldots, R_m$. Let $D$ be the observed failure times before arriving at the predefined time $T$.

The GPHC scheme can be classified into the following categories:

Case 1: $X_1, \ldots, X_d, \ldots, X_k$ for $T < X_k < X_m$,
Case 2: $X_1, \ldots, X_k, \ldots, X_d$ for $X_k < T < X_m$,
Case 3: $X_1, \ldots, X_k, \ldots, X_m$ for $X_k < X_m < T$,

According to the GPHC scheme, the joint density function for three different cases is as follows:

$$f_X(x) = A^* \prod_{i=1}^D f(x_{i;m;n})[\bar{F}(x_{i;m;n})]^{R_i^*}[\bar{F}(T)]^{R_\tau^*}, \tag{4}$$

where

$$D = \begin{cases} k \text{ If } T < X_{k,m,n} < X_{m,m,n} \\ d \text{ If } X_{k,m,n} < T < X_{m,m,n} \\ m \text{ If} X_{k,m,n} < X_{m,m,n} < T \end{cases}, \qquad A^* = \prod_{i=1}^{D} \sum_{j=i}^{m} (R_j^* + 1),$$

$$R^* = \begin{cases} (R_1, \ldots, R_d, 0, \ldots, 0, R_k^* = n - k - \sum_{j=1}^{d} R_j) & \text{If } T < X_{k,m,n} < X_{m,m,n} \\ (R_1, \ldots, R_d) & \text{If } X_{k,m,n} < T < X_{m,m,n} \\ (R_1, \ldots, R_m) & \text{If} X_{k,m,n} < X_{m,m,n} < T \end{cases}$$

with $R_\tau^*$ the number of surviving units that are removed at $T$, given by

$$R_\tau^* = \begin{cases} 0 & \text{If } T < X_{k,m,n} < X_{m,m,n} \\ n - d - \sum_{j=1}^{d} R_j & \text{If } X_{k,m,n} < T < X_{m,m,n} \\ 0 & \text{If} X_{k,m,n} < X_{m,m,n} < T \end{cases}.$$

## 3. Maximum Likelihood Estimation

The ML procedure is a popular and effective strategy used by statisticians when dealing with reliability issues and survival analysis. This section provides the MLE of $R$ and the approximate CI is established.

### 3.1. Maximum Likelihood Estimator of R

Here, the MLE of $R$ is determined, first by obtaining the MLEs of $\alpha_1, \alpha_2, \alpha_3,$ and $\lambda$. Plugging the PDF and CDF of the EED, i.e., (1) and (2), into the likelihood formula (4), the likelihood function of $\alpha_1, \alpha_2, \alpha_3,$ and $\lambda$ is expressed as:

$$l(\alpha_1, \alpha_2, \alpha_3, \lambda) = A_1^* A_2^* A_3^* \alpha_1^{D_1} \alpha_2^{D_2} \alpha_3^{D_3} \lambda^{D_1+D_2+D_3}$$

$$\times \prod_{j=1}^{D_1} e^{-\lambda x_j} (1 - e^{-\lambda x_j})^{\alpha_1 - 1} (1 - (1 - e^{-\lambda x_j})^{\alpha_1})^{R_{j1}} (1 - (1 - e^{-\lambda T_1})^{\alpha_1})^{R_{d1+1}^*}$$

$$\times \prod_{j=1}^{D_2} e^{-\lambda y_j} (1 - e^{-\lambda y_j})^{\alpha_2 - 1} (1 - (1 - e^{-\lambda y_j})^{\alpha_2})^{R_{j2}} (1 - (1 - e^{-\lambda T_2})^{\alpha_2})^{R_{d2+1}^*}$$

$$\times \prod_{j=1}^{D_3} e^{-\lambda z_j} (1 - e^{-\lambda z_j})^{\alpha_3 - 1} (1 - (1 - e^{-\lambda z_j})^{\alpha_3})^{R_{j3}} (1 - (1 - e^{-\lambda T_3})^{\alpha_3})^{R_{d3+1}^*}. \tag{5}$$

Taking the logarithm for (5), we obtain the log-likelihood function

$$L = \ln l(\alpha_1, \alpha_2, \alpha_3, \lambda) = \ln A_1^* A_2^* A_3^* + D_1 \ln \alpha_1 + D_2 \ln \alpha_2 + D_3 \ln \alpha_3 + (D_1 + D_2 + D_3) \ln \lambda$$

$$- \lambda \left[ \sum_{j=1}^{D_1} x_j + \sum_{j=1}^{D_2} y_j + \sum_{j=1}^{D_3} z_j \right] + (\alpha_1 - 1) \sum_{j=1}^{D_1} \ln \psi_1(x_j, \lambda) + (\alpha_2 - 1) \sum_{j=1}^{D_2} \ln \psi_2(y_j, \lambda)$$

$$+ (\alpha_3 - 1) \sum_{j=1}^{D_3} \ln \psi_3(z_j, \lambda) + \sum_{j=1}^{D_1} R_{j1} \ln(1 - (\psi_1(x_j, \lambda))^{\alpha_1}) + \sum_{j=1}^{D_2} R_{j2} \ln(1 - (\psi_2(y_j, \lambda))^{\alpha_2})$$

$$+ \sum_{j=1}^{D_3} R_{j3} \ln(1 - (\psi_3(z_j, \lambda))^{\alpha_3}) + R_{d1+1}^* \ln(1 - (\psi_1(T_1, \lambda))^{\alpha_1}) + R_{d2+1}^* \ln(1 - (\psi_2(T_2, \lambda))^{\alpha_2})$$

$$+ R_{d3+1}^* \ln(1 - (\psi_3(T_3, \lambda))^{\alpha_3}) \tag{6}$$

where

$$\begin{cases} \psi_1(x_j,\lambda) = 1 - e^{-\lambda x_j}, \\ \psi_2(y_j,\lambda) = 1 - e^{-\lambda y_j}, \\ \psi_3(z_j,\lambda) = 1 - e^{-\lambda z_j}, \end{cases} \tag{7}$$

and $\psi_1(T_1,\lambda)$ is as given by (7) with $x_j = T_1$. Similarly for $\psi_2(T_2,\lambda)$ and $\psi_3(T_3,\lambda)$. We take the partial derivatives of (6) for $\alpha_1, \alpha_2, \alpha_3,$ and $\lambda$, respectively, and obtain a set of likelihood equations as follows:

$$\begin{cases} \dfrac{\partial L}{\partial \alpha_1} = \dfrac{D_1}{\alpha_1} + \displaystyle\sum_{j=1}^{D_1}[\ln\psi_1(x_j,\lambda) - R_{j1}E_1(\psi_1(x_j,\lambda),\alpha_1)] - R^*_{d1+1}E_1(\psi_1(T_1,\lambda),\alpha_1), \\[2mm] \dfrac{\partial L}{\partial \alpha_2} = \dfrac{D_2}{\alpha_2} + \displaystyle\sum_{j=1}^{D_2}[\ln\psi_2(y_j,\lambda) - R_{j2}E_2(\psi_2(y_j,\lambda),\alpha_2)] - R^*_{d2+1}E_2(\psi_2(T_2,\lambda),\alpha_2), \\[2mm] \dfrac{\partial L}{\partial \alpha_3} = \dfrac{D_3}{\alpha_3} + \displaystyle\sum_{j=1}^{D_3}[\ln\psi_3(z_j,\lambda) - R_{j3}E_3(\psi_3(z_j,\lambda),\alpha_3)] - R^*_{d3+1}E_3(\psi_3(T_3,\lambda),\alpha_3), \\[2mm] \dfrac{\partial L}{\partial \lambda} = \dfrac{D_1+D_2+D_3}{\lambda} - [\displaystyle\sum_{j=1}^{D_1}x_j + \sum_{j=1}^{D_2}y_j + \sum_{j=1}^{D_3}z_j] + (\alpha_1-1)\sum_{j=1}^{D_1}Q_1(\psi_1(x_j,\lambda),\lambda) + (\alpha_2-1)\sum_{j=1}^{D_2}Q_2(\psi_2(y_j,\lambda),\lambda) \\[2mm] \quad + (\alpha_3-1)\displaystyle\sum_{j=1}^{D_3}Q_3(\psi_3(z_j,\lambda),\lambda) - \sum_{j=1}^{D_1}R_{j1}S_1(\psi_1(x_j,\lambda),\lambda) - \sum_{j=1}^{D_2}R_{j2}S_2(\psi_2(y_j,\lambda),\lambda) - \sum_{j=1}^{D_3}R_{j3}S_3(\psi_3(z_j,\lambda),\lambda) \\[2mm] \quad - R^*_{d1+1}S_1(\psi_1(T_1,\lambda),\lambda) - R^*_{d2+1}S_2(\psi_2(T_2,\lambda),\lambda) - R^*_{d3+1}S_3(\psi_3(T_3,\lambda),\lambda), \end{cases} \tag{8}$$

where

$$E_i(\psi_i(w_j,\lambda),\alpha_i) = \frac{(\psi_i(w_j,\lambda))^{\alpha_i}}{1-(\psi_i(w_j,\lambda))^{\alpha_i}}\ln\psi_i(w_j,\lambda),$$

$$E_i(\psi_i(T_i,\lambda),\alpha_i) = \frac{(\psi_i(T_i,\lambda))^{\alpha_i}}{1-(\psi_i(T_i,\lambda))^{\alpha_i}}\ln\psi_i(T_i,\lambda),$$

$$Q_i(\psi_i(w_j,\lambda),\lambda) = \frac{w_j e^{-\lambda w_j}}{\psi_i(w_j,\lambda)},$$

$$S_i(\psi_i(w_j,\lambda),\lambda) = \frac{\alpha_i(\psi_i(w_j,\lambda))^{\alpha_i}}{1-(\psi_i(w_j,\lambda))^{\alpha_i}}Q_i(\psi_i(w_j,\lambda),\lambda),$$

where $w = x, y, z, i = 1, 2, 3$.

There are no analytical solutions for the unknown parameters $\alpha_1, \alpha_2, \alpha_3,$ and $\lambda$ from (8). As a result, (8) may be maximized using a fairly straightforward iterative approach such as the Newton–Raphson (NR) procedure to obtain the appropriate MLEs of $\alpha_1, \alpha_2, \alpha_3,$ and $\lambda$. The corresponding precise distribution (or exact confidence intervals (CIs) ) of $\alpha_1, \alpha_2, \alpha_3,$ and $\lambda$ cannot be obtained because the MLEs of $\alpha_1, \alpha_2, \alpha_3,$ and $\lambda$ cannot be obtained in closed form. For this purpose, we recommend implementing the 'maxLik' package for any given dataset $(\alpha_1, \alpha_2, \alpha_3, \lambda)$ and then performing ML calculations using the NR iterative method. We can substitute these estimates to obtain the MLE of $R$ as:

$$\hat{R} = \frac{\hat{\alpha}_2\hat{\alpha}_3}{(\hat{\alpha}_1+\hat{\alpha}_2)(\hat{\alpha}_1+\hat{\alpha}_2+\hat{\alpha}_3)}. \tag{9}$$

### 3.2. Asymptotic Confidence Interval

CIs, which contain the population parameter with a given high probability, show the level of uncertainty in an estimate derived from sample data. Utilizing the large-sample normal distribution of the MLE is the most popular technique for establishing confidence bounds for the parameters. For a large sample size, MLEs are consistent and asymptotically normally distributed.

Here, the asymptotic CI of $R$ is determined using the asymptotic distribution of $\hat{R}$, which was obtained from the asymptotic distribution of $\alpha_1, \alpha_2, \alpha_3$, and $\lambda$. The observed Fisher information matrix is denoted by

$$\mathbf{I}(\theta) = [\mathbf{I}_{ij}] = \left[ -\frac{\partial^2 L}{\partial \theta_i \partial \theta_j} \right], i, j = 1, 2, 3.$$

The elements in lines of the $\mathbf{I}(\theta)$ matrix can be obtained by differentiating from (8) two times with respect to $\alpha_1, \alpha_2, \alpha_3$, and $\lambda$:

$$I_{11} = -\left[ \frac{D_1}{\alpha_1^2} + \sum_{j=1}^{D_1} R_{j1} \frac{\partial E_1(\psi_1(x_j, \lambda), \alpha_1)}{\partial \alpha_1} + R_{d1+1}^* \frac{\partial E_1(\psi_1(T_1, \lambda), \alpha_1)}{\partial \alpha_1} \right],$$

$$I_{22} = -\left[ \frac{D_2}{\alpha_2^2} + \sum_{j=1}^{D_2} R_{j2} \frac{\partial E_2(\psi_2(y_j, \lambda), \alpha_2)}{\partial \alpha_2} + R_{d2+1}^* \frac{\partial E_2(\psi_2(T_2, \lambda), \alpha_2)}{\partial \alpha_2} \right],$$

$$I_{33} = -\left[ \frac{D_3}{\alpha_3^2} + \sum_{j=1}^{D_3} R_{j3} \frac{\partial E_3(\psi_3(z_j, \lambda), \alpha_3)}{\partial \alpha_3} + R_{d3+1}^* \frac{\partial E_3(\psi_3(T_3, \lambda), \alpha_3)}{\partial \alpha_3} \right],$$

$$I_{14} = \sum_{j=1}^{D_1} \left[ Q_1(\psi_1(x_j, \lambda) - R_{j1} \frac{\partial E_1(\psi_1(x_j, \lambda), \alpha_1)}{\partial \lambda} \right] - R_{d1+1}^* \frac{\partial E_1(\psi_1(T_1, \lambda), \alpha_1)}{\partial \lambda},$$

$$I_{24} = \sum_{j=1}^{D_2} \left[ Q_2(\psi_2(y_j, \lambda) - R_{j2} \frac{\partial E_2(\psi_2(y_j, \lambda), \alpha_2)}{\partial \lambda} \right] - R_{d2+1}^* \frac{\partial E_2(\psi_2(T_2, \lambda), \alpha_2)}{\partial \lambda},$$

$$I_{34} = \sum_{j=1}^{D_3} \left[ Q_3(\psi_3(z_j, \lambda) - R_{j3} \frac{\partial E_3(\psi_3(z_j, \lambda), \alpha_3)}{\partial \lambda} \right] - R_{d3+1}^* \frac{\partial E_3(\psi_3(T_3, \lambda), \alpha_3)}{\partial \lambda},$$

$$I_{44} = -\frac{D_1 + D_2 + D_3}{\lambda^2} + (\alpha_1 - 1) \sum_{j=1}^{D_1} \frac{\partial Q_1(\psi_1(x_j, \lambda), \lambda)}{\partial \lambda} + (\alpha_2 - 1) \sum_{j=1}^{D_2} \frac{\partial Q_2(\psi_2(y_j, \lambda), \lambda)}{\partial \lambda},$$

$$+ (\alpha_3 - 1) \sum_{j=1}^{D_3} \frac{\partial Q_3(\psi_3(z_j, \lambda), \lambda)}{\partial \lambda} - \sum_{j=1}^{D_1} R_{j1} \frac{\partial S_1(\psi_1(x_j, \lambda), \lambda)}{\partial \lambda} - \sum_{j=1}^{D_2} R_{j2} \frac{\partial S_2(\psi_2(y_j, \lambda), \lambda)}{\partial \lambda},$$

$$- \sum_{j=1}^{D_3} R_{j3} \frac{\partial S_3(\psi_3(z_j, \lambda), \lambda)}{\partial \lambda} - R_{d1+1}^* \frac{\partial S_1(\psi_1(T_1, \lambda), \lambda)}{\partial \lambda} -$$

$$R_{d2+1}^* \frac{\partial S_2(\psi_2(T_2, \lambda), \lambda)}{\partial \lambda} - R_{d3+1}^* \frac{\partial S_3(\psi_3(T_3, \lambda), \lambda)}{\partial \lambda},$$

$$I_{12} = I_{21} = I_{13} = I_{31} = I_{32} = I_{23} = 0,$$

where

$$\frac{\partial E_i(\psi_i(w_j, \lambda), \alpha_i)}{\partial \alpha_i} = \frac{(E_i(\psi_i(w_j, \lambda), \alpha_i))^2}{(\psi_i(w_j, \lambda))^{\alpha_i}},$$

$$\frac{\partial E_i(\psi_i(T_i, \lambda), \alpha_i)}{\partial \alpha_i} = \frac{(E_i(\psi_i(T_i, \lambda), \alpha_i))^2}{(\psi_i(T_i, \lambda))^{\alpha_i}},$$

$$\frac{\partial E_i(\psi_i(w_j, \lambda), \alpha_i)}{\partial \lambda} = S_i(\psi_i(w_j, \lambda), \lambda)[\frac{1}{\alpha_i} + \ln \psi_i(w_j, \lambda) + E_i(\psi_i(w_j, \lambda), \alpha_i)],$$

$$\frac{\partial Q_i(\psi_i(w_j, \lambda), \lambda)}{\partial \lambda} = -Q_i(\psi_i(w_j, \lambda), \lambda)((w_j + Q_i(\psi_i(w_j, \lambda), \lambda)),$$

$$\frac{\partial S_i(\psi_i(w_j, \lambda), \lambda)}{\partial \lambda} = S_i(\psi_i(w_j, \lambda), \lambda)[(\alpha_i - 1)Q_i(\psi_i(w_j, \lambda), \lambda) + S_i(\psi_i(w_j, \lambda), \lambda) - w_j],$$

where $w = x, y, z, i = 1, 2, 3$.

As $D_1 \to \infty, D_2 \to \infty, D_3 \to \infty$, then

$$[\sqrt{n_1}(\hat{\alpha}_1 - \alpha_1), \sqrt{n_2}(\hat{\alpha}_2 - \alpha_2), \sqrt{n_3}(\hat{\alpha}_3 - \alpha_3)] \to N(0, I^{-1}(\alpha_1, \alpha_2, \alpha_3)),$$

where

$$I^{-1}(\alpha_1, \alpha_2, \alpha_3) = \begin{pmatrix} \frac{1}{I_{11}} & 0 & 0 \\ 0 & \frac{1}{I_{22}} & 0 \\ 0 & 0 & \frac{1}{I_{33}} \end{pmatrix}.$$

The asymptotic distribution of $\hat{R}$ is found as

$$[(\hat{R} - R)] \to N(0, \sigma_R^2),$$

where $\sigma_R^2 = M^T I^{-1} M$ is the asymptotic variance of $R$, and

$$M = \left( \frac{\partial R}{\partial \alpha_1}, \frac{\partial R}{\partial \alpha_2}, \frac{\partial R}{\partial \alpha_3} \right),$$

where

$$\frac{\partial R}{\partial \alpha_1} = \frac{-\alpha_2 \alpha_3 (2(\alpha_1 + \alpha_2) + \alpha_3)}{(\alpha_1 + \alpha_2)^2 (\alpha_1 + \alpha_2 + \alpha_3)^2},$$

$$\frac{\partial R}{\partial \alpha_2} = \frac{-\alpha_3 (\alpha_1 (\alpha_1 + \alpha_3) - \alpha_2^2)}{(\alpha_1 + \alpha_2)^2 (\alpha_1 + \alpha_2 + \alpha_3)^2},$$

$$\frac{\partial R}{\partial \alpha_3} = \frac{\alpha_2}{(\alpha_1 + \alpha_2 + \alpha_3)^2}.$$

Therefore, a $100(1 - \gamma)\%$ asymptotic CI of $R$ can be constructed as,

$$(\hat{R} - z_{1-\frac{\gamma}{2}} \sqrt{\hat{\sigma}_R^2}, \hat{R} + z_{1-\frac{\gamma}{2}} \sqrt{\hat{\sigma}_R^2}),$$

where $z_{\frac{\gamma}{2}}$ is the $100\gamma$ th standard normal percentile and $\hat{\sigma}_R$ is the standard deviation of the MLE of $\hat{R}$.

## 4. Bayesian Estimation

In this section, the Bayesian estimation of $R$ is obtained when data are observed using the GPHC based on the squared error loss function (SELF) and linear exponential (Linex) loss function, which are defined, respectively, by

$$L_1 = (\vartheta, \check{\vartheta}) = (\check{\vartheta} - \vartheta)^2,$$

$$L_2 = (\vartheta, \check{\vartheta}) = e^{g(\check{\vartheta} - \vartheta)} - g(\check{\vartheta} - \vartheta) - 1,$$

where $\check{\vartheta}$ is an estimator of $\vartheta$. Denote the prior and posterior distributions of $\vartheta$ by $\pi(\vartheta)$ and $\pi^*(\vartheta \mid \underline{x})$, respectively. Under the SELF and Linex loss function, the Bayesian estimation of any function $B(\vartheta)$ of $\vartheta$ is given by

$$B(\vartheta)_{SE} = E[B(\vartheta) \mid \underline{x}] = \int_0^\infty B(\vartheta) \pi^*(\vartheta \mid \underline{x}) d\vartheta,$$

$$B(\vartheta)_{LN} = \frac{-1}{c} \ln\left( E(e^{-cB(\vartheta)}) \right) = \frac{-1}{c} \ln\left( \int_0^\infty e^{-cB(\vartheta)} \pi^*(\vartheta \mid \underline{x}) d\vartheta \right).$$

Prior distribution is important for the development of Bayes estimators.

Under the assumption of gamma prior distributions, we investigate at this estimate problem. The gamma family of distributions is particularly adaptable and can be thought of as an appropriate prior for any unknown parameter(s) because it has a wide variety of shapes depending on its parameter values; for further information, see references Kundu [42] and Dey et al. [43]. Furthermore, the independent gamma priors are simple and transparent, which may help to avoid many challenging inferential problems.

Therefore, it is assumed here that $\alpha_1, \alpha_2, \alpha_3,$ and $\lambda$ follow independent gamma distributions with $\alpha_1 \sim G(\eta_1, \zeta_1), \alpha_2 \sim G(\eta_2, \zeta_2), \alpha_3 \sim G(\eta_3, \zeta_3),$ and $\lambda \sim G(\eta_4, \zeta_4),$ with probability densities given by, respectively,

$$\pi(\alpha_i) = \frac{\alpha_i^{\eta_i-1}}{\Gamma(\eta_i)\zeta_i^{\eta_i}} e^{-\frac{\alpha_i}{\zeta_i}}, \quad \pi(\lambda) = \frac{\lambda^{\eta_4-1}}{\Gamma(\eta_4)\zeta_4^{\eta_4}} e^{-\frac{\lambda}{\zeta_4}}, \quad \alpha_i > 0, \eta_i, \zeta_i > 0, i = 1, 2, 3. \tag{10}$$

Using the informative prior (10) and the likelihood function (5), the joint posterior density can be derived as follows:

$$\pi^*(\alpha_1, \alpha_2, \alpha_3, \lambda) = \prod_{i=1}^{3} A_i^* \frac{\alpha_i^{D_i+\eta_i-1}}{\Gamma(\eta_i)\zeta_i^{\eta_i}} \frac{\lambda^{\eta_4-1+\sum_{i=1}^{3} D_i}}{\Gamma(\eta_4)\zeta_4^{\eta_4}} e^{-\frac{\lambda}{\zeta_4}}$$

$$\times e^{-\sum_{i=1}^{D_1}\left[\lambda x_i + \frac{\alpha_1}{\zeta_1} - (\alpha_1-1)\ln\psi_1(x_i,\lambda) - R_{i1}\ln(1-(\psi_1(x_i,\lambda))^{\alpha_1})\right] + R_{d1+1}^*\ln(1-(\psi_1(T_1,\lambda))^{\alpha_1})}$$

$$\times e^{-\sum_{i=1}^{D_2}\left[\lambda y_i + \frac{\alpha_2}{\zeta_2} - (\alpha_2-1)\ln\psi_2(y_i,\lambda) - R_{i2}\ln(1-(\psi_2(y_i,\lambda))^{\alpha_2})\right] + R_{d2+1}^*\ln(1-(\psi_2(T_2,\lambda))^{\alpha_2})}$$

$$\times e^{-\sum_{i=1}^{D_3}\left[\lambda z_i + \frac{\alpha_3}{\zeta_3} - (\alpha_3-1)\ln\psi_3(z_i,\lambda) - R_{i3}\ln(1-(\psi_3(z_i,\lambda))^{\alpha_3})\right] + R_{d3+1}^*\ln(1-(\psi_3(T_3,\lambda))^{\alpha_3})}.$$

Then, the Bayes estimates of $R$ under the SELF and Linex loss function, say $\tilde{R}_{SE}$ and $\tilde{R}_{LN}$ are given by

$$\begin{cases} \tilde{R}_{SE} = \int_0^\infty \int_0^\infty \int_0^\infty R\pi^*(\alpha_1, \alpha_2, \alpha_3, \lambda) d\alpha_1 d\alpha_2 d\alpha_3, \\ \tilde{R}_{LN} = \frac{-1}{c} \int_0^\infty \int_0^\infty \int_0^\infty e^{-cR}\pi^*(\alpha_1, \alpha_2, \alpha_3, \lambda) d\alpha_1 d\alpha_2 d\alpha_3. \end{cases} \tag{11}$$

Obviously, it is not possible to compute (11) analytically. The MCMC approaches can be applied to approximate (11).

The marginal posterior densities of the parameters $\alpha_1, \alpha_2, \alpha_3,$ and $\lambda$ can be derived as

$$\begin{cases} \pi^*(\alpha_1) \propto \alpha_1^{D_1+\eta_1-1} e^{-\sum_{i=1}^{D_1}\left[\alpha_1(\frac{1}{\zeta_1}-\ln\psi_1(x_i,\lambda)) - R_{i1}\ln(1-(\psi_1(x_i,\lambda))^{\alpha_1})\right] + R_{d1+1}^*\ln(1-(\psi_1(T_1,\lambda))^{\alpha_1})} \\ \pi^*(\alpha_2) \propto \alpha_2^{D_2+\eta_2-1} e^{-\sum_{i=1}^{D_2}\left[\alpha_2(\frac{1}{\zeta_2}-\ln\psi_2(y_i,\lambda)) - R_{i2}\ln(1-(\psi_2(y_i,\lambda))^{\alpha_2})\right] + R_{d2+1}^*\ln(1-(\psi_2(T_2,\lambda))^{\alpha_2})} \\ \pi^*(\alpha_3) \propto \alpha_3^{D_3+\eta_3-1} e^{-\sum_{i=1}^{D_3}\left[\alpha_3(\frac{1}{\zeta_3}-\ln\psi_3(z_i,\lambda)) - R_{i3}\ln(1-(\psi_3(z_i,\lambda))^{\alpha_3})\right] + R_{d3+1}^*\ln(1-(\psi_3(T_3,\lambda))^{\alpha_3})} \\ \pi^*(\lambda) \propto \lambda^{\eta_4-1+\sum_{i=1}^{3} D_i} e^{-\frac{\lambda}{\zeta_4}} e^{-\sum_{i=1}^{D_1}[\lambda x_i - (\alpha_1-1)\ln\psi_1(x_i,\lambda) - R_{i1}\ln(1-(\psi_1(x_i,\lambda))^{\alpha_1})] + R_{d1+1}^*\ln(1-(\psi_1(T_1,\lambda))^{\alpha_1})} \\ \quad \times e^{-\sum_{i=1}^{D_2}[\lambda y_i - (\alpha_2-1)\ln\psi_2(y_i,\lambda) - R_{i2}\ln(1-(\psi_2(y_i,\lambda))^{\alpha_2})] + R_{d2+1}^*\ln(1-(\psi_2(T_2,\lambda))^{\alpha_2})} \\ \quad \times e^{-\sum_{i=1}^{D_3}[\lambda z_i - (\alpha_3-1)\ln\psi_3(z_i,\lambda) - R_{i3}\ln(1-(\psi_3(z_i,\lambda))^{\alpha_3})] + R_{d3+1}^*\ln(1-(\psi_3(T_3,\lambda))^{\alpha_3})}, \end{cases} \tag{12}$$

The marginal posterior densities in (12) are not well-known distributions, so we will use the Metropolis–Hastings (MH) sampler to generate the values of $\alpha_1, \alpha_2, \alpha_3,$ and $\lambda$ with normal proposal distribution to generate samples from it in (12)

Furthermore, the approach of Chen and Shao [44] was extensively used to construct HPD intervals with unknown benefit distribution parameters for Bayesian estimate. For example, a 95% HPD interval can be created using two endpoints from the MCMC sample outputs: 2.5% and 97.5% percentiles, respectively. The $\Theta$ parameters' Bayes, trustworthy intervals are calculated as follows:

1.  Sorted parameters as $\tilde{\alpha}_l^{[1]} < \tilde{\alpha}_l^{[2]} < \ldots < \tilde{\alpha}_l^{[N]}$; $l = 1, 2, 3$, $\tilde{\lambda}^{[1]} < \tilde{\lambda}^{[2]} < \ldots < \tilde{\lambda}^{[N]}$, and $R^{[1]} < R^{[2]} < \ldots < R^{[N]}$, and $N$ is the length of MCMC generated.

2.  The 95% symmetric credible intervals of $\tilde{\alpha}_1, \tilde{\alpha}_2, \tilde{\alpha}_3, \tilde{\lambda}$, and $\tilde{R}$ become $\left( \tilde{\alpha}_l^{L\frac{25}{1000}}, \tilde{\alpha}_l^{L\frac{975}{1000}} \right)$, $\left( \tilde{\lambda}^{L\frac{25}{1000}}, \tilde{\lambda}^{L\frac{975}{1000}} \right)$, and $\left( \tilde{R}^{L\frac{25}{1000}}, \tilde{R}^{L\frac{975}{1000}} \right)$.

## 5. Simulation Study

We give some simulation results in this section to show how the various strategies presented in this paper perform in practice. Different situations have been used as:

Case I: $\alpha_1 = 0.5, \alpha_2 = 4, \alpha_3 = 20, \lambda = 0.5$ and $T_1 = 14, T_2 = 6, T_3 = 18$.
Case II: $\alpha_1 = 0.8, \alpha_2 = 5, \alpha_3 = 12, \lambda = 1.5$ and $T_1 = 1.6, T_2 = 2, T_3 = 2.5$.
Case III: $\alpha_1 = 0.3, \alpha_2 = 2, \alpha_3 = 15, \lambda = 0.5$ and $T_1 = 3.5, T_2 = 3, T_3 = 6$.
Case IV: $\alpha_1 = 0.7, \alpha_2 = 3, \alpha_3 = 8, \lambda = 2$ and $T_1 = 1, T_2 = 1.5, T_3 = 1.9$.

We have considered different sample sizes ($n$) for each sample as $n_1 = 20, n_2 = 25, n_3 = 15$, and $n_1 = 30, n_2 = 40, n_3 = 30$; different effective sample sizes ($m$) for each sample as $m_1 = 15, m_2 = 18, m_3 = 12$, and $m_1 = 17, m_2 = 22, m_3 = 14$; different $k$ values for each sample as $k_1 = 12, k_2 = 16, k_3 = 10$, and $k_1 = 15, k_2 = 20, k_3 = 12$; and two different progressive censoring schemes, namely Scheme-I $R_1 = (n_1 - m_1, rep(0, m_1 - 1))$, $R_2 = (n_2 - m_2, rep(0, m_2 - 1))$, $R_3 = (n_3 - m_3, rep(0, m_3 - 1))$ and Scheme-II $R_1 = (2, rep(0, (m_1 - 3)/2), n_1 - m_1 - 3, rep(0, (m_1 - 3)/2), 1)$, $R_2 = (1, rep(0, (m_2 - 2)/2), n_2 - m_2 - 2, rep(0, (m_2 - 4)/2), 1)$, $R_3 = (1, rep(0, (m_3 - 3)/2), n_3 - m_3 - 2, rep(0, (m_3 - 1)/2), 1)$, where *rep* points to replicate censored scheme. In each case, we compute and construct the 95% exact and the credible CIs for both loss functions. The methods are repeated 5000 times, and the average estimators (AvE), mean squared errors (MSE), average length of CIs (L.CI) with related coverage percentages (CP), and average length of HPD credible CIs (L.CCI) are reported.

For prior distribution, the hyperparameters are chosen using elective hyperparameters based on mean and variance of gamma prior distribution.

To find out how to elicit hyperparameters of the independent joint prior, we can utilise the likelihood method's estimate and variance–covariance matrix. The mean and variance of gamma priors can be used to represent the derived hyperparameters.

$$\eta_j = \frac{\left[ \frac{1}{N} \sum_{i=1}^{N} \hat{\vartheta}_j^i \right]^2}{\frac{1}{N-1} \sum_{i=1}^{N} \left[ \hat{\vartheta}_j^i - \frac{1}{N} \sum_{i=1}^{N} \hat{\vartheta}_j^i \right]^2}; \ j = 1, \ldots, p,$$

$$\zeta_j = \frac{\frac{1}{N} \sum_{i=1}^{N} \hat{\vartheta}_j^i}{\frac{1}{N-1} \sum_{i=1}^{N} \left[ \hat{\vartheta}_j^i - \frac{1}{N} \sum_{i=1}^{N} \hat{\vartheta}_j^i \right]^2}; \ j = 1, \ldots, p,$$

where $N$ is the number of simulation iterations.

For MCMC techniques, we replicate the process 10,000 times of MH algorithms. Figures 2 and 3 are explained by Table 1 that discusses different label of these figures as MSE.i.1 is MLE, MSE.i.2 is SELF, MSE.i.3 is Linex c = −0.5, and MSE.i.4 is Linex c = 1.5, where $i = 1, \ldots, 8$. The X label indicates different parameters as param1 is $\alpha_1$, param2 is $\alpha_2$, param3 is $\alpha_3$, and param4 is $\lambda$. On the left side of Tables 2–5, the average of the frequentist estimates and the average of the Bayesian MCMC estimates of $\alpha_1, \alpha_2, \alpha_3$, and $\lambda$ are determined. Additionally, on the right side of Tables 2–5, we report the L.CI and L.CCI of $\alpha_1, \alpha_2, \alpha_3$, and $\lambda$ that were obtained. Two highly regarded R software packages are used to conduct extensive evaluations: the 'CODA' package for computing Bayes estimates using MCMC techniques and the 'maxLik' tool for computing ML estimates using the NR algorithm.

The heatmap plots of the MSE of simulation results for parameters and $R$ of Cases I, II, III, and IV are provided in Figures 2 and 3, respectively. Using one variable on each axis,

heatmaps are used to display relationships between two variables (MSE and methods). You can determine if there are any trends in the values for one or both variables by monitoring how cell colors vary across each axis. MSE was represented by cell coloring, with the darker color representing a higher MSE value than the lighter color. The results are presented in Tables 2–5, and the following observations can be made:

- ML and Bayesian estimates of population parameters are quite good based on AvE where they tend to actual values.
- As the sample size increases, the MSE decreases as expected for ML and Bayesian estimations.
- For a given sample size, the MSE also declines with $m$.
- If $n$ and $m$ are held constant, the MSEs decrease as the acceptable bare minimum of failures, $k$, rises.
- Based on AvE, MSE, and length of CI, we note scheme II is better than scheme I in same times.
- Bayesian estimates outperform ML estimates in terms of AvE, MSE, and length of CI because they incorporate prior knowledge based on a gamma informative prior.
- When the symmetric and asymmetric loss functions are compared, Bayes estimates under the asymmetric loss function are found to be more accurate than others under the symmetric loss function.
- The average length of HPD credible intervals for Bayesian estimation is better than length of asymptotic confidence interval.

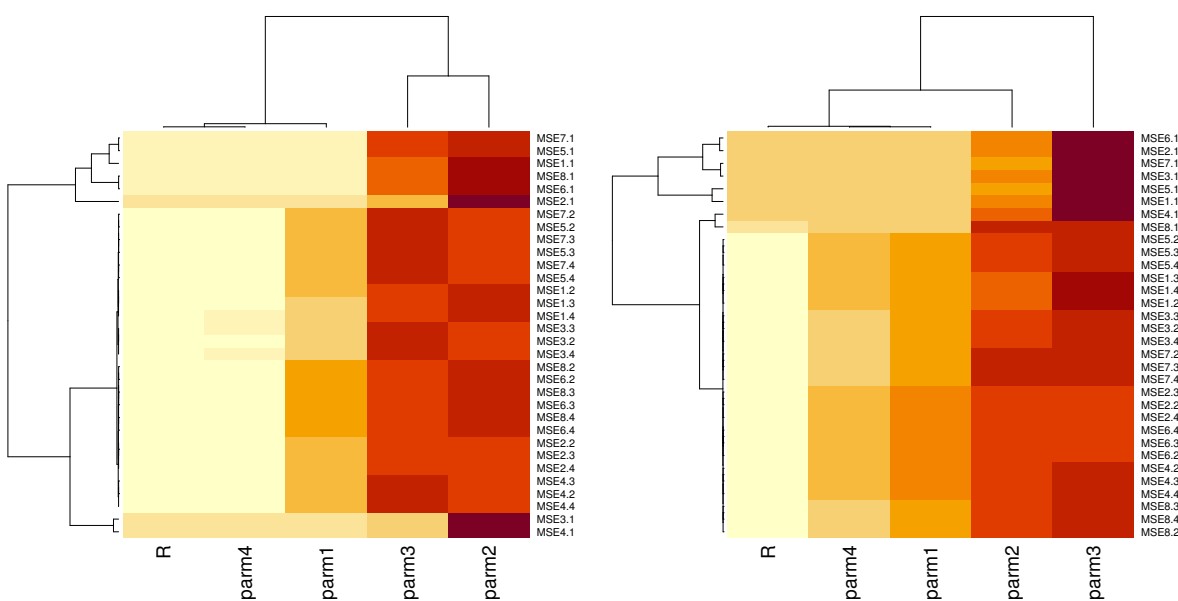

**Figure 2.** Heatmaps of MSE for parameters and R based on different estimation methods: Case I and II.

**Table 1.** Labels of MSE results for heatmaps plots.

| scheme | | 1 | | | |
|---|---|---|---|---|---|
| $n_1, n_2, n_3$ | | 20, 25, 15 | | 30, 40, 30 | |
| $m_1, m_2, m_3$ | 15, 18, 12 | 17, 22, 14 | 25, 30, 22 | 27, 35, 26 | |
| $k_1, k_2, k_3$ | 12, 16, 10 | 15, 20, 12 | 20, 24, 20 | 23, 30, 23 | |
| label | MSE1 | MSE2 | MSE3 | MSE4 | |
| scheme | | 2 | | | |
| $n_1, n_2, n_3$ | | 20, 25, 15 | | 30, 40, 30 | |
| $m_1, m_2, m_3$ | 15, 18, 12 | 17, 22, 14 | 25, 30, 22 | 27, 35, 26 | |
| $k_1, k_2, k_3$ | 12, 16, 10 | 15, 20, 12 | 20, 24, 20 | 23, 30, 23 | |
| label | MSE5 | MSE6 | MSE7 | MSE8 | |

**Table 2.** AvE and MSE for estimation methods: Case I.

| | | | | | $\alpha_1 = 0.5, \alpha_2 = 4, \alpha_3 = 20, \lambda = 0.5$ | | | | | | | | | | | | |
|---|---|---|---|---|---|---|---|---|---|---|---|---|---|---|---|---|---|
| | $T_1 = 14, T_2 = 6, T_3 = 18$ | | | | ML | | SELF | | Linex c = −0.5 | | Linex c = 1.5 | | ML | | SELF | c = −0.5 | c = 1.5 |
| Scheme | $n_1, n_2, n_3$ | $m_1, m_2, m_3$ | $k_1, k_2, k_3$ | | AvE | MSE | AvE | MSE | AvE | MSE | AvE | MSE | L.CI | CP | L.CCI | L.CCI | L.CCI |
| 1 | 20, 25, 15 | 15, 18, 12 | 12, 16, 10 | $\alpha_1$ | 0.5378 | 0.0204 | 0.5073 | 0.0098 | 0.5054 | 0.0097 | 0.5016 | 0.0095 | 0.5401 | 95.80% | 0.3781 | 0.3699 | 0.3701 |
| | | | | $\alpha_2$ | 4.1354 | 0.8059 | 3.9970 | 0.0234 | 3.9942 | 0.0234 | 3.9887 | 0.0235 | 3.4822 | 94.40% | 0.5716 | 0.5715 | 0.5701 |
| | | | | $\alpha_3$ | 20.0513 | 0.5460 | 19.9954 | 0.0216 | 19.9926 | 0.0218 | 19.9872 | 0.0221 | 2.8925 | 95.80% | 0.5676 | 0.5639 | 0.5553 |
| | | | | $\lambda$ | 0.5058 | 0.0021 | 0.5075 | 0.0019 | 0.5069 | 0.0018 | 0.5057 | 0.0018 | 0.1776 | 94.40% | 0.1584 | 0.1583 | 0.1581 |
| | | | | R | 0.7153 | 0.0009 | 0.7245 | 0.0004 | 0.7249 | 0.0003 | 0.7256 | 0.0003 | 0.1089 | 95.80% | 0.0728 | 0.0728 | 0.0723 |
| | | 17, 22, 14 | 15, 20, 12 | $\alpha_1$ | 0.5311 | 0.0176 | 0.5016 | 0.0072 | 0.5004 | 0.0071 | 0.4981 | 0.0071 | 0.5059 | 95.20% | 0.3151 | 0.3154 | 0.3159 |
| | | | | $\alpha_2$ | 4.0615 | 0.7533 | 4.0026 | 0.0125 | 4.0011 | 0.0125 | 3.9980 | 0.0125 | 3.3972 | 95.80% | 0.4228 | 0.4221 | 0.4160 |
| | | | | $\alpha_3$ | 20.0648 | 0.2245 | 19.9961 | 0.0128 | 19.9945 | 0.0128 | 19.9912 | 0.0129 | 1.8419 | 95.20% | 0.4168 | 0.4173 | 0.4176 |
| | | | | $\lambda$ | 0.5010 | 0.0017 | 0.5039 | 0.0014 | 0.5035 | 0.0014 | 0.5026 | 0.0014 | 0.1627 | 95.80% | 0.1465 | 0.1465 | 0.1460 |
| | | | | R | 0.7174 | 0.0008 | 0.7255 | 0.0003 | 0.7258 | 0.0003 | 0.7262 | 0.0003 | 0.1038 | 95.20% | 0.0613 | 0.0612 | 0.0612 |
| | 30, 40, 30 | 25, 30, 22 | 20, 24, 20 | $\alpha_1$ | 0.5151 | 0.0112 | 0.5053 | 0.0070 | 0.5037 | 0.0070 | 0.5005 | 0.0068 | 0.4113 | 95.40% | 0.3228 | 0.3207 | 0.3156 |
| | | | | $\alpha_2$ | 4.0874 | 0.4318 | 4.0056 | 0.0192 | 4.0030 | 0.0191 | 3.9977 | 0.0190 | 2.5557 | 95.60% | 0.5481 | 0.5483 | 0.5488 |
| | | | | $\alpha_3$ | 20.0091 | 0.0337 | 19.9921 | 0.0220 | 19.9894 | 0.0221 | 19.9841 | 0.0223 | 0.7193 | 95.40% | 0.5763 | 0.5751 | 0.5755 |
| | | | | $\lambda$ | 0.5019 | 0.0010 | 0.5033 | 0.0009 | 0.5030 | 0.0009 | 0.5023 | 0.0009 | 0.1244 | 95.60% | 0.1134 | 0.1135 | 0.1138 |
| | | | | R | 0.7208 | 0.0004 | 0.7247 | 0.0003 | 0.7251 | 0.0003 | 0.7257 | 0.0002 | 0.0807 | 95.40% | 0.0618 | 0.0609 | 0.0595 |
| | | 27, 35, 26 | 23, 30, 23 | $\alpha_1$ | 0.5248 | 0.0103 | 0.5128 | 0.0067 | 0.5117 | 0.0067 | 0.5095 | 0.0065 | 0.3852 | 95.20% | 0.3092 | 0.3057 | 0.3035 |
| | | | | $\alpha_2$ | 4.0518 | 0.3875 | 3.9927 | 0.0130 | 3.9911 | 0.0130 | 3.9879 | 0.0131 | 2.4343 | 95.40% | 0.4393 | 0.4430 | 0.4463 |
| | | | | $\alpha_3$ | 20.0208 | 0.0305 | 19.9952 | 0.0150 | 19.9935 | 0.0150 | 19.9901 | 0.0152 | 0.6884 | 95.20% | 0.4650 | 0.4643 | 0.4668 |
| | | | | $\lambda$ | 0.4999 | 0.0008 | 0.5018 | 0.0008 | 0.5015 | 0.0008 | 0.5009 | 0.0008 | 0.1127 | 95.40% | 0.1071 | 0.1070 | 0.1068 |
| | | | | R | 0.7194 | 0.0004 | 0.7235 | 0.0002 | 0.7237 | 0.0002 | 0.7241 | 0.0002 | 0.0761 | 95.20% | 0.0604 | 0.0602 | 0.0593 |
| 2 | 20, 25, 15 | 15, 18, 12 | 12, 16, 10 | $\alpha_1$ | 0.5235 | 0.0155 | 0.5064 | 0.0105 | 0.5047 | 0.0104 | 0.5012 | 0.0102 | 0.4792 | 94.40% | 0.3828 | 0.3819 | 0.3814 |
| | | | | $\alpha_2$ | 4.1142 | 0.7423 | 4.0001 | 0.0215 | 3.9975 | 0.0215 | 3.9923 | 0.0215 | 3.3510 | 95.20% | 0.5642 | 0.5629 | 0.5569 |
| | | | | $\alpha_3$ | 20.0442 | 0.5969 | 20.0032 | 0.0238 | 20.0004 | 0.0238 | 19.9948 | 0.0238 | 3.0265 | 94.40% | 0.5783 | 0.5787 | 0.5797 |
| | | | | $\lambda$ | 0.5018 | 0.0018 | 0.5029 | 0.0016 | 0.5024 | 0.0016 | 0.5013 | 0.0016 | 0.1648 | 95.20% | 0.1544 | 0.1545 | 0.1547 |
| | | | | R | 0.7187 | 0.0007 | 0.7248 | 0.0004 | 0.7251 | 0.0004 | 0.7258 | 0.0004 | 0.0993 | 94.40% | 0.0736 | 0.0736 | 0.0734 |
| | | 17, 22, 14 | 15, 20, 12 | $\alpha_1$ | 0.5302 | 0.0148 | 0.5000 | 0.0084 | 0.4988 | 0.0083 | 0.4964 | 0.0082 | 0.4519 | 94.80% | 0.3470 | 0.3459 | 0.3470 |
| | | | | $\alpha_2$ | 4.1304 | 0.5943 | 4.0028 | 0.0142 | 4.0011 | 0.0142 | 3.9976 | 0.0142 | 2.9815 | 94.40% | 0.4777 | 0.4773 | 0.4754 |
| | | | | $\alpha_3$ | 20.0581 | 0.4443 | 20.0036 | 0.0126 | 20.0019 | 0.0126 | 19.9984 | 0.0126 | 2.6054 | 94.80% | 0.4490 | 0.4477 | 0.4501 |
| | | | | $\lambda$ | 0.5026 | 0.0015 | 0.5054 | 0.0015 | 0.5049 | 0.0015 | 0.5040 | 0.0015 | 0.1525 | 94.40% | 0.1414 | 0.1411 | 0.1411 |
| | | | | R | 0.7174 | 0.0006 | 0.7259 | 0.0003 | 0.7262 | 0.0003 | 0.7266 | 0.0003 | 0.0910 | 94.80% | 0.0661 | 0.0659 | 0.0664 |
| | 30, 40, 30 | 25, 30, 22 | 20, 24, 20 | $\alpha_1$ | 0.5235 | 0.0155 | 0.5064 | 0.0105 | 0.5047 | 0.0104 | 0.5012 | 0.0102 | 0.4792 | 94.40% | 0.3828 | 0.3819 | 0.3814 |
| | | | | $\alpha_2$ | 4.1142 | 0.7423 | 4.0001 | 0.0215 | 3.9975 | 0.0215 | 3.9923 | 0.0215 | 3.3510 | 95.20% | 0.5642 | 0.5629 | 0.5569 |
| | | | | $\alpha_3$ | 20.0442 | 0.5969 | 20.0032 | 0.0238 | 20.0004 | 0.0238 | 19.9948 | 0.0238 | 3.0265 | 94.40% | 0.5783 | 0.5787 | 0.5797 |
| | | | | $\lambda$ | 0.5018 | 0.0018 | 0.5029 | 0.0016 | 0.5024 | 0.0016 | 0.5013 | 0.0016 | 0.1648 | 95.20% | 0.1544 | 0.1545 | 0.1547 |
| | | | | R | 0.7187 | 0.0007 | 0.7248 | 0.0004 | 0.7251 | 0.0004 | 0.7258 | 0.0004 | 0.0993 | 94.40% | 0.0736 | 0.0736 | 0.0734 |
| | | 27, 35, 26 | 23, 30, 23 | $\alpha_1$ | 0.5302 | 0.0148 | 0.5000 | 0.0084 | 0.4988 | 0.0083 | 0.4964 | 0.0082 | 0.4519 | 94.80% | 0.3470 | 0.3459 | 0.3470 |
| | | | | $\alpha_2$ | 4.1304 | 0.5943 | 4.0028 | 0.0142 | 4.0011 | 0.0142 | 3.9976 | 0.0142 | 2.9815 | 94.40% | 0.4777 | 0.4773 | 0.4754 |
| | | | | $\alpha_3$ | 20.0581 | 0.4443 | 20.0036 | 0.0126 | 20.0019 | 0.0126 | 19.9984 | 0.0126 | 2.6054 | 94.80% | 0.4490 | 0.4477 | 0.4501 |
| | | | | $\lambda$ | 0.5026 | 0.0015 | 0.5054 | 0.0015 | 0.5049 | 0.0015 | 0.5040 | 0.0015 | 0.1525 | 94.40% | 0.1414 | 0.1411 | 0.1411 |
| | | | | R | 0.7174 | 0.0006 | 0.7259 | 0.0003 | 0.7262 | 0.0003 | 0.7266 | 0.0003 | 0.0910 | 94.80% | 0.0661 | 0.0659 | 0.0664 |

**Table 3.** AvE and MSE for estimation methods: Case II.

| | | | | | $\alpha_1 = 0.8, \alpha_2 = 5, \alpha_3 = 12, \lambda = 1.5$ | | | | | | | | | | | | |
| --- | --- | --- | --- | --- | --- | --- | --- | --- | --- | --- | --- | --- | --- | --- | --- | --- | --- |
| | $T_1=1.6, T_2=2, T_3=2.5$ | | | | ML | | SELF | | Linex c = −0.5 | | Linex c = 1.5 | | ML | | SELF | c = −0.5 | c = 1.5 |
| Scheme | $n_1,n_2,n_3$ | $m_1,m_2,m_3$ | $k_1,k_2,k_3$ | | AvE | MSE | AvE | MSE | AvE | MSE | AvE | MSE | L.CI | CP | L.CCI | L.CCI | L.CCI |
| 1 | 20, 25, 15 | 15, 18, 12 | 12, 16, 10 | $\alpha_1$ | 0.8559 | 0.0569 | 0.8005 | 0.0120 | 0.7984 | 0.0119 | 0.7943 | 0.0119 | 0.9098 | 95.60% | 0.4097 | 0.4119 | 0.4129 |
| | | | | $\alpha_2$ | 5.4973 | 2.8046 | 4.9991 | 0.0184 | 4.9964 | 0.0184 | 4.9911 | 0.0184 | 6.2750 | 95.20% | 0.5209 | 0.5208 | 0.5209 |
| | | | | $\alpha_3$ | 12.7108 | 8.2755 | 12.0099 | 0.0257 | 12.0069 | 0.0255 | 12.0009 | 0.0253 | 12.2652 | 95.60% | 0.6073 | 0.6075 | 0.6097 |
| | | | | $\lambda$ | 1.5320 | 0.0322 | 1.5022 | 0.0088 | 1.5004 | 0.0087 | 1.4969 | 0.0087 | 0.6931 | 95.20% | 0.3507 | 0.3479 | 0.3437 |
| | | | | R | 0.5697 | 0.0032 | 0.5815 | 0.0002 | 0.5818 | 0.0002 | 0.5823 | 0.0002 | 0.2174 | 95.60% | 0.0555 | 0.0555 | 0.0561 |
| | | 17, 22, 14 | 15, 20, 12 | $\alpha_1$ | 0.8511 | 0.0482 | 0.7976 | 0.0098 | 0.7963 | 0.0098 | 0.7936 | 0.0098 | 0.8379 | 95.20% | 0.3794 | 0.3799 | 0.3837 |
| | | | | $\alpha_2$ | 5.5710 | 1.8904 | 4.9993 | 0.0141 | 4.9976 | 0.0141 | 4.9941 | 0.0142 | 6.2835 | 96.00% | 0.4370 | 0.4389 | 0.4459 |
| | | | | $\alpha_3$ | 12.9478 | 5.2061 | 12.0025 | 0.0144 | 12.0008 | 0.0144 | 11.9974 | 0.0144 | 10.3100 | 95.20% | 0.4541 | 0.4543 | 0.4611 |
| | | | | $\lambda$ | 1.5417 | 0.0316 | 1.4997 | 0.0076 | 1.4985 | 0.0076 | 1.4961 | 0.0076 | 0.6729 | 96.00% | 0.3286 | 0.3295 | 0.3314 |
| | | | | R | 0.5738 | 0.0024 | 0.5818 | 0.0002 | 0.5820 | 0.0002 | 0.5823 | 0.0002 | 0.1914 | 95.20% | 0.0520 | 0.0521 | 0.0523 |
| | 30, 40, 30 | 25, 30, 22 | 20, 24, 20 | $\alpha_1$ | 0.8229 | 0.0308 | 0.8107 | 0.0126 | 0.8087 | 0.0125 | 0.8047 | 0.0122 | 0.6824 | 95.00% | 0.4215 | 0.4232 | 0.4261 |
| | | | | $\alpha_2$ | 5.2341 | 1.2492 | 4.9958 | 0.0200 | 4.9932 | 0.0200 | 4.9880 | 0.0202 | 4.2884 | 95.00% | 0.5420 | 0.5400 | 0.5509 |
| | | | | $\alpha_3$ | 12.2260 | 3.7063 | 11.9974 | 0.0212 | 11.9946 | 0.0213 | 11.9891 | 0.0214 | 7.5020 | 95.00% | 0.5633 | 0.5657 | 0.5616 |
| | | | | $\lambda$ | 1.5096 | 0.0175 | 1.5054 | 0.0069 | 1.5040 | 0.0069 | 1.5010 | 0.0068 | 0.5181 | 95.00% | 0.3188 | 0.3182 | 0.3167 |
| | | | | R | 0.5746 | 0.0016 | 0.5800 | 0.0002 | 0.5803 | 0.0002 | 0.5808 | 0.0002 | 0.1527 | 95.00% | 0.0560 | 0.0554 | 0.0552 |
| | | 27, 35, 26 | 23, 30, 23 | $\alpha_1$ | 0.8350 | 0.0298 | 0.8013 | 0.0089 | 0.8000 | 0.0088 | 0.7975 | 0.0088 | 0.6634 | 95.40% | 0.3535 | 0.3516 | 0.3475 |
| | | | | $\alpha_2$ | 5.3222 | 0.9419 | 5.0056 | 0.0122 | 5.0039 | 0.0122 | 5.0006 | 0.0122 | 4.1500 | 95.40% | 0.4298 | 0.4321 | 0.4286 |
| | | | | $\alpha_3$ | 12.2497 | 1.7810 | 11.9989 | 0.0143 | 11.9973 | 0.0143 | 11.9940 | 0.0144 | 6.5668 | 95.40% | 0.4754 | 0.4728 | 0.4689 |
| | | | | $\lambda$ | 1.5176 | 0.0168 | 1.5040 | 0.0059 | 1.5029 | 0.0059 | 1.5007 | 0.0059 | 0.5025 | 95.40% | 0.2907 | 0.2918 | 0.2929 |
| | | | | R | 0.5723 | 0.0013 | 0.5811 | 0.0002 | 0.5812 | 0.0002 | 0.5816 | 0.0002 | 0.1394 | 95.40% | 0.0482 | 0.0478 | 0.0480 |
| 2 | 20, 25, 15 | 15, 18, 12 | 12, 16, 10 | $\alpha_1$ | 0.8742 | 0.0514 | 0.8041 | 0.0133 | 0.8021 | 0.0132 | 0.7982 | 0.0131 | 0.8402 | 95.60% | 0.4472 | 0.4478 | 0.4468 |
| | | | | $\alpha_2$ | 5.4432 | 2.3905 | 4.9916 | 0.0199 | 4.9891 | 0.0199 | 4.9840 | 0.0201 | 5.8123 | 95.40% | 0.5518 | 0.5563 | 0.5516 |
| | | | | $\alpha_3$ | 12.9312 | 9.1114 | 12.0084 | 0.0244 | 12.0055 | 0.0243 | 11.9996 | 0.0243 | 11.2667 | 95.60% | 0.6050 | 0.6027 | 0.6014 |
| | | | | $\lambda$ | 1.5469 | 0.0323 | 1.4974 | 0.0096 | 1.4957 | 0.0096 | 1.4922 | 0.0096 | 0.6804 | 95.40% | 0.3929 | 0.3924 | 0.3888 |
| | | | | R | 0.5730 | 0.0023 | 0.5812 | 0.0003 | 0.5814 | 0.0003 | 0.5820 | 0.0002 | 0.1862 | 95.60% | 0.0639 | 0.0637 | 0.0636 |
| | | 17, 22, 14 | 15, 20, 12 | $\alpha_1$ | 0.8560 | 0.0501 | 0.7986 | 0.0102 | 0.7972 | 0.0101 | 0.7945 | 0.0101 | 0.8350 | 94.80% | 0.3881 | 0.3887 | 0.3886 |
| | | | | $\alpha_2$ | 5.3897 | 2.1790 | 4.9994 | 0.0129 | 4.9978 | 0.0129 | 4.9947 | 0.0129 | 5.5868 | 94.20% | 0.4366 | 0.4356 | 0.4377 |
| | | | | $\alpha_3$ | 12.6666 | 5.1642 | 12.0019 | 0.0136 | 12.0003 | 0.0135 | 11.9970 | 0.0135 | 9.5871 | 94.80% | 0.4329 | 0.4348 | 0.4344 |
| | | | | $\lambda$ | 1.5270 | 0.0270 | 1.5003 | 0.0064 | 1.4991 | 0.0064 | 1.4968 | 0.0064 | 0.6357 | 94.20% | 0.3010 | 0.3024 | 0.3051 |
| | | | | R | 0.5718 | 0.0022 | 0.5816 | 0.0002 | 0.5818 | 0.0002 | 0.5822 | 0.0002 | 0.1790 | 94.80% | 0.0533 | 0.0528 | 0.0525 |
| | 30, 40, 30 | 25, 30, 22 | 20, 24, 20 | $\alpha_1$ | 0.8469 | 0.0309 | 0.8056 | 0.0111 | 0.8036 | 0.0110 | 0.7997 | 0.0109 | 0.6650 | 95.40% | 0.3944 | 0.3935 | 0.3911 |
| | | | | $\alpha_2$ | 5.2695 | 1.1245 | 4.9884 | 0.0200 | 4.9856 | 0.0201 | 4.9802 | 0.0204 | 4.0244 | 94.80% | 0.5569 | 0.5573 | 0.5509 |
| | | | | $\alpha_3$ | 12.2875 | 3.8795 | 11.9932 | 0.0213 | 11.9903 | 0.0214 | 11.9845 | 0.0215 | 7.6459 | 95.40% | 0.5772 | 0.5785 | 0.5787 |
| | | | | $\lambda$ | 1.5197 | 0.0150 | 1.5078 | 0.0069 | 1.5063 | 0.0068 | 1.5034 | 0.0068 | 0.4748 | 94.80% | 0.3096 | 0.3092 | 0.3105 |
| | | | | R | 0.5722 | 0.0014 | 0.5807 | 0.0002 | 0.5810 | 0.0002 | 0.5815 | 0.0002 | 0.1413 | 95.40% | 0.0535 | 0.0536 | 0.0540 |
| | | 27, 35, 26 | 23, 30, 23 | $\alpha_1$ | 0.8382 | 0.0302 | 0.8075 | 0.0083 | 0.8062 | 0.0082 | 0.8037 | 0.0081 | 0.6581 | 94.60% | 0.3504 | 0.3463 | 0.3435 |
| | | | | $\alpha_2$ | 5.2528 | 1.1020 | 5.0078 | 0.0125 | 5.0061 | 0.0124 | 5.0027 | 0.0124 | 3.9272 | 95.60% | 0.4439 | 0.4424 | 0.4393 |
| | | | | $\alpha_3$ | 12.1636 | 1.1223 | 11.9998 | 0.0140 | 11.9981 | 0.0140 | 11.9946 | 0.0140 | 6.7565 | 94.60% | 0.4503 | 0.4444 | 0.4456 |
| | | | | $\lambda$ | 1.5169 | 0.0146 | 1.4989 | 0.0052 | 1.4979 | 0.0052 | 1.4958 | 0.0052 | 0.4693 | 95.60% | 0.2851 | 0.2861 | 0.2842 |
| | | | | R | 0.5761 | 0.0013 | 0.5802 | 0.0002 | 0.5804 | 0.0001 | 0.5807 | 0.0001 | 0.1377 | 94.60% | 0.0477 | 0.0474 | 0.0469 |

**Table 4.** AvE and MSE for estimation methods: Case III.

| | | | | | | | | | | | | | | | | | | |
|---|---|---|---|---|---|---|---|---|---|---|---|---|---|---|---|---|---|---|
| | | | | | \multicolumn{9}{} $\alpha_1 = 0.3, \alpha_2 = 2, \alpha_3 = 15, \lambda = 0.5$ | | | | | | | | | |
| | $T_1 = 3.5, T_2 = 3, T_3 = 6$ | | | | ML | | SELF | | Linex c = −0.5 | | Linex c = 1.5 | | ML | | SELF | c = −0.5 | c = 1.5 | |
| Scheme | $n_1, n_2, n_3$ | $m_1, m_2, m_3$ | $k_1, k_2, k_3$ | | AvE | MSE | AvE | MSE | AvE | MSE | AvE | MSE | L.CI | CP | L.CCI | L.CCI | L.CCI |
| 1 | 20, 25, 15 | 15, 18, 12 | 12, 16, 10 | $\alpha_1$ | 0.3166 | 0.0071 | 0.3148 | 0.0068 | 0.3134 | 0.0067 | 0.3106 | 0.0064 | 0.3232 | 95.40% | 0.2974 | 0.2950 | 0.2912 |
| | | | | $\alpha_2$ | 2.1113 | 0.2775 | 1.9928 | 0.0193 | 1.9902 | 0.0194 | 1.9851 | 0.0197 | 2.0204 | 96.20% | 0.5380 | 0.5395 | 0.5349 |
| | | | | $\alpha_3$ | 15.1804 | 0.9295 | 15.0086 | 0.0237 | 15.0057 | 0.0235 | 14.9999 | 0.0233 | 3.7162 | 95.40% | 0.5848 | 0.5859 | 0.5892 |
| | | | | $\lambda$ | 0.5089 | 0.0035 | 0.5081 | 0.0028 | 0.5072 | 0.0027 | 0.5055 | 0.0027 | 0.2298 | 96.20% | 0.1908 | 0.1904 | 0.1895 |
| | | | | R | 0.7461 | 0.0010 | 0.7491 | 0.0009 | 0.7495 | 0.0009 | 0.7504 | 0.0009 | 0.1202 | 95.40% | 0.1097 | 0.1096 | 0.1087 |
| | | 17, 22, 14 | 15, 20, 12 | $\alpha_1$ | 0.3119 | 0.0056 | 0.3061 | 0.0044 | 0.3051 | 0.0043 | 0.3032 | 0.0043 | 0.2887 | 96.20% | 0.2471 | 0.2460 | 0.2461 |
| | | | | $\alpha_2$ | 2.1496 | 0.2802 | 2.0004 | 0.0125 | 1.9989 | 0.0125 | 1.9958 | 0.0125 | 1.9925 | 95.20% | 0.4500 | 0.4494 | 0.4458 |
| | | | | $\alpha_3$ | 15.0655 | 0.6932 | 14.9944 | 0.0133 | 14.9927 | 0.0133 | 14.9893 | 0.0135 | 3.2570 | 96.20% | 0.4373 | 0.4400 | 0.4415 |
| | | | | $\lambda$ | 0.5055 | 0.0029 | 0.5021 | 0.0025 | 0.5015 | 0.0025 | 0.5002 | 0.0025 | 0.2087 | 95.20% | 0.1977 | 0.1976 | 0.1959 |
| | | | | R | 0.7467 | 0.0008 | 0.7522 | 0.0006 | 0.7525 | 0.0006 | 0.7531 | 0.0006 | 0.1067 | 96.20% | 0.0907 | 0.0901 | 0.0897 |
| | 30, 40, 30 | 25, 30, 22 | 20, 24, 20 | $\alpha_1$ | 0.3118 | 0.0040 | 0.3139 | 0.0043 | 0.3128 | 0.0042 | 0.3108 | 0.0041 | 0.2435 | 95.20% | 0.2401 | 0.2388 | 0.2377 |
| | | | | $\alpha_2$ | 2.0837 | 0.1850 | 2.0001 | 0.0205 | 1.9975 | 0.0205 | 1.9923 | 0.0205 | 1.6556 | 95.40% | 0.5507 | 0.5522 | 0.5557 |
| | | | | $\alpha_3$ | 15.1172 | 1.1520 | 15.0028 | 0.0217 | 14.9999 | 0.0217 | 14.9941 | 0.0218 | 4.1865 | 95.20% | 0.5453 | 0.5459 | 0.5549 |
| | | | | $\lambda$ | 0.5020 | 0.0018 | 0.5022 | 0.0014 | 0.5016 | 0.0014 | 0.5006 | 0.0014 | 0.1686 | 95.40% | 0.1489 | 0.1484 | 0.1467 |
| | | | | R | 0.7481 | 0.0006 | 0.7491 | 0.0006 | 0.7494 | 0.0006 | 0.7501 | 0.0005 | 0.0924 | 95.20% | 0.0886 | 0.0884 | 0.0886 |
| | | 27, 35, 26 | 23, 30, 23 | $\alpha_1$ | 0.3081 | 0.0039 | 0.3039 | 0.0036 | 0.3032 | 0.0035 | 0.3016 | 0.0035 | 0.2432 | 94.40% | 0.2244 | 0.2237 | 0.2223 |
| | | | | $\alpha_2$ | 2.0500 | 0.1520 | 2.0008 | 0.0122 | 1.9992 | 0.0122 | 1.9959 | 0.0123 | 1.5171 | 95.20% | 0.4266 | 0.4288 | 0.4296 |
| | | | | $\alpha_3$ | 15.0618 | 0.9047 | 15.0049 | 0.0128 | 15.0033 | 0.0127 | 15.0001 | 0.0127 | 3.7243 | 94.40% | 0.4193 | 0.4192 | 0.4187 |
| | | | | $\lambda$ | 0.4987 | 0.0015 | 0.5014 | 0.0014 | 0.5010 | 0.0014 | 0.5002 | 0.0014 | 0.1518 | 95.20% | 0.1503 | 0.1505 | 0.1510 |
| | | | | R | 0.7493 | 0.0006 | 0.7529 | 0.0005 | 0.7531 | 0.0005 | 0.7537 | 0.0005 | 0.0931 | 94.40% | 0.0824 | 0.0825 | 0.0821 |
| 2 | 20, 25, 15 | 15, 18, 12 | 12, 16, 10 | $\alpha_1$ | 0.3189 | 0.0071 | 0.3121 | 0.0047 | 0.3108 | 0.0046 | 0.3084 | 0.0045 | 0.2983 | 95.80% | 0.2414 | 0.2399 | 0.2371 |
| | | | | $\alpha_2$ | 2.0828 | 0.2477 | 1.9838 | 0.0222 | 1.9809 | 0.0224 | 1.9752 | 0.0227 | 1.9258 | 95.40% | 0.5688 | 0.5659 | 0.5681 |
| | | | | $\alpha_3$ | 15.1029 | 0.5541 | 14.9962 | 0.0203 | 14.9935 | 0.0203 | 14.9880 | 0.0204 | 2.2510 | 95.80% | 0.5643 | 0.5609 | 0.5610 |
| | | | | $\lambda$ | 0.5042 | 0.0029 | 0.5056 | 0.0024 | 0.5048 | 0.0023 | 0.5033 | 0.0023 | 0.2101 | 95.40% | 0.1847 | 0.1842 | 0.1828 |
| | | | | R | 0.7441 | 0.0009 | 0.7496 | 0.0006 | 0.7500 | 0.0006 | 0.7508 | 0.0006 | 0.1105 | 95.80% | 0.0924 | 0.0925 | 0.0925 |
| | | 17, 22, 14 | 15, 20, 12 | $\alpha_1$ | 0.3180 | 0.0061 | 0.3018 | 0.0046 | 0.3009 | 0.0045 | 0.2990 | 0.0044 | 0.2322 | 94.60% | 0.2471 | 0.2469 | 0.2440 |
| | | | | $\alpha_2$ | 2.1092 | 0.2276 | 2.0018 | 0.0136 | 2.0001 | 0.0136 | 1.9968 | 0.0136 | 1.8222 | 94.60% | 0.4706 | 0.4693 | 0.4651 |
| | | | | $\alpha_3$ | 15.0212 | 0.3295 | 14.9973 | 0.0137 | 14.9956 | 0.0137 | 14.9923 | 0.0137 | 2.0893 | 94.60% | 0.4469 | 0.4477 | 0.4434 |
| | | | | $\lambda$ | 0.5030 | 0.0025 | 0.5036 | 0.0023 | 0.5029 | 0.0024 | 0.5016 | 0.0024 | 0.1974 | 94.60% | 0.1927 | 0.1921 | 0.1904 |
| | | | | R | 0.7456 | 0.0009 | 0.7538 | 0.0006 | 0.7541 | 0.0006 | 0.7547 | 0.0006 | 0.1017 | 94.60% | 0.0919 | 0.0921 | 0.0917 |
| | 30, 40, 30 | 25, 30, 22 | 20, 24, 20 | $\alpha_1$ | 0.3107 | 0.0037 | 0.3096 | 0.0035 | 0.3087 | 0.0034 | 0.3067 | 0.0034 | 0.2366 | 95.80% | 0.2227 | 0.2219 | 0.2201 |
| | | | | $\alpha_2$ | 2.0320 | 0.1328 | 1.9892 | 0.0221 | 1.9866 | 0.0222 | 1.9815 | 0.0226 | 1.4245 | 95.20% | 0.5694 | 0.5654 | 0.5668 |
| | | | | $\alpha_3$ | 15.1021 | 0.7021 | 14.9879 | 0.0204 | 14.9853 | 0.0205 | 14.9801 | 0.0207 | 3.2634 | 95.80% | 0.5299 | 0.5313 | 0.5318 |
| | | | | $\lambda$ | 0.5005 | 0.0014 | 0.5027 | 0.0012 | 0.5022 | 0.0012 | 0.5013 | 0.0012 | 0.1474 | 95.20% | 0.1336 | 0.1333 | 0.1332 |
| | | | | R | 0.7489 | 0.0006 | 0.7502 | 0.0005 | 0.7505 | 0.0005 | 0.7512 | 0.0005 | 0.0901 | 95.80% | 0.0820 | 0.0817 | 0.0806 |
| | | 27, 35, 26 | 23, 30, 23 | $\alpha_1$ | 0.3117 | 0.0038 | 0.3086 | 0.0035 | 0.3078 | 0.0034 | 0.3062 | 0.0033 | 0.2368 | 95.60% | 0.2151 | 0.2136 | 0.2131 |
| | | | | $\alpha_2$ | 2.0803 | 0.1250 | 2.0017 | 0.0120 | 2.0002 | 0.0120 | 1.9972 | 0.0120 | 1.4086 | 96.00% | 0.4076 | 0.4062 | 0.4030 |
| | | | | $\alpha_3$ | 15.0494 | 0.4042 | 14.9978 | 0.0131 | 14.9961 | 0.0131 | 14.9928 | 0.0132 | 2.4872 | 95.60% | 0.4394 | 0.4412 | 0.4450 |
| | | | | $\lambda$ | 0.5065 | 0.0014 | 0.5055 | 0.0013 | 0.5051 | 0.0012 | 0.5043 | 0.0013 | 0.1437 | 96.00% | 0.1406 | 0.1406 | 0.1401 |
| | | | | R | 0.7482 | 0.0005 | 0.7510 | 0.0005 | 0.7513 | 0.0005 | 0.7519 | 0.0005 | 0.0882 | 95.60% | 0.0805 | 0.0792 | 0.0776 |

**Table 5.** AvE and MSE for estimation methods: Case IV.

| | | | | | $\alpha_1 = 0.7, \alpha_2 = 3, \alpha_3 = 8, \lambda = 2$ | | | | | | | | | | | | |
| | $T_1 = 1, T_2 = 1.5, T_3 = 1.9$ | | | | ML | | SELF | | Linex c = −0.5 | | Linex c = 1.5 | | ML | | SELF | c = −0.5 | c = 1.5 |
| Scheme | $n_1,n_2,n_3$ | $m_1,m_2,m_3$ | $k_1,k_2,k_3$ | | AvE | MSELF | AvE | MSE | AvE | MSE | AvE | MSE | L.CI | CP | L.CCI | L.CCI | L.CCI |
|---|---|---|---|---|---|---|---|---|---|---|---|---|---|---|---|---|---|
| 1 | 20, 25, 15 | 15, 18, 12 | 12, 16, 10 | $\alpha_1$ | 0.7635 | 0.0417 | 0.7033 | 0.0123 | 0.7014 | 0.0123 | 0.6975 | 0.0122 | 0.7611 | 95.00% | 0.4289 | 0.4289 | 0.4266 |
| | | | | $\alpha_2$ | 3.3870 | 1.3437 | 3.0005 | 0.0210 | 2.9978 | 0.0210 | 2.9925 | 0.0211 | 4.2876 | 95.80% | 0.5508 | 0.5480 | 0.5441 |
| | | | | $\alpha_3$ | 9.3616 | 9.3014 | 7.9996 | 0.0227 | 7.9967 | 0.0227 | 7.9910 | 0.0228 | 10.2762 | 95.00% | 0.5867 | 0.5911 | 0.5934 |
| | | | | $\lambda$ | 2.0807 | 0.1086 | 2.0015 | 0.0137 | 1.9994 | 0.0136 | 1.9953 | 0.0135 | 1.2539 | 95.80% | 0.4473 | 0.4472 | 0.4465 |
| | | | | R | 0.5523 | 0.0033 | 0.5542 | 0.0005 | 0.5546 | 0.0005 | 0.5553 | 0.0005 | 0.2251 | 95.00% | 0.0855 | 0.0851 | 0.0846 |
| | | 17, 22, 14 | 15, 20, 12 | $\alpha_1$ | 0.7647 | 0.0404 | 0.7022 | 0.0092 | 0.7008 | 0.0092 | 0.6980 | 0.0091 | 0.7185 | 96.00% | 0.3731 | 0.3725 | 0.3693 |
| | | | | $\alpha_2$ | 3.4365 | 1.0365 | 2.9949 | 0.0129 | 2.9933 | 0.0129 | 2.9901 | 0.0130 | 4.2520 | 94.80% | 0.4327 | 0.4321 | 0.4262 |
| | | | | $\alpha_3$ | 9.3034 | 7.5958 | 7.9983 | 0.0145 | 7.9966 | 0.0145 | 7.9933 | 0.0145 | 9.3445 | 96.00% | 0.4664 | 0.4658 | 0.4637 |
| | | | | $\lambda$ | 2.1014 | 0.1045 | 1.9955 | 0.0091 | 1.9942 | 0.0091 | 1.9917 | 0.0091 | 1.2043 | 94.80% | 0.3827 | 0.3830 | 0.3834 |
| | | | | R | 0.5507 | 0.0031 | 0.5544 | 0.0004 | 0.5546 | 0.0004 | 0.5551 | 0.0004 | 0.2131 | 96.00% | 0.0735 | 0.0733 | 0.0726 |
| | 30, 40, 30 | 25, 30, 22 | 20, 24, 20 | $\alpha_1$ | 0.7336 | 0.0233 | 0.7089 | 0.0124 | 0.7069 | 0.0123 | 0.7028 | 0.0120 | 0.5846 | 95.40% | 0.4094 | 0.4129 | 0.4123 |
| | | | | $\alpha_2$ | 3.2429 | 0.6714 | 2.9939 | 0.0213 | 2.9913 | 0.0214 | 2.9861 | 0.0215 | 3.0707 | 94.60% | 0.5588 | 0.5627 | 0.5669 |
| | | | | $\alpha_3$ | 8.8894 | 6.9282 | 7.9936 | 0.0239 | 7.9908 | 0.0239 | 7.9852 | 0.0240 | 9.7209 | 95.40% | 0.6159 | 0.6138 | 0.6050 |
| | | | | $\lambda$ | 2.0719 | 0.0705 | 2.0018 | 0.0107 | 1.9999 | 0.0107 | 1.9961 | 0.0106 | 1.0027 | 94.60% | 0.4126 | 0.4123 | 0.4110 |
| | | | | R | 0.5550 | 0.0021 | 0.5531 | 0.0005 | 0.5534 | 0.0005 | 0.5542 | 0.0005 | 0.1794 | 95.40% | 0.0821 | 0.0821 | 0.0823 |
| | | 27, 35, 26 | 23, 30, 23 | $\alpha_1$ | 0.7285 | 0.0228 | 0.7030 | 0.0085 | 0.7017 | 0.0085 | 0.6991 | 0.0084 | 0.5824 | 95.80% | 0.3447 | 0.3438 | 0.3420 |
| | | | | $\alpha_2$ | 3.1671 | 0.4801 | 3.0004 | 0.0147 | 2.9987 | 0.0147 | 2.9953 | 0.0147 | 2.6385 | 95.40% | 0.4820 | 0.4778 | 0.4715 |
| | | | | $\alpha_3$ | 8.8246 | 6.8463 | 8.0027 | 0.0138 | 8.0009 | 0.0137 | 7.9975 | 0.0137 | 9.7439 | 95.80% | 0.4601 | 0.4616 | 0.4614 |
| | | | | $\lambda$ | 2.0596 | 0.0542 | 2.0090 | 0.0077 | 2.0077 | 0.0077 | 2.0051 | 0.0076 | 0.8830 | 95.40% | 0.3385 | 0.3401 | 0.3398 |
| | | | | R | 0.5561 | 0.0020 | 0.5542 | 0.0003 | 0.5545 | 0.0003 | 0.5549 | 0.0003 | 0.1762 | 95.80% | 0.0693 | 0.0694 | 0.0691 |
| 2 | 20, 25, 15 | 15, 18, 12 | 12, 16, 10 | $\alpha_1$ | 0.7424 | 0.0440 | 0.6994 | 0.0140 | 0.6972 | 0.0139 | 0.6928 | 0.0138 | 0.8062 | 95.00% | 0.4481 | 0.4468 | 0.4455 |
| | | | | $\alpha_2$ | 3.3167 | 1.1318 | 2.9927 | 0.0231 | 2.9898 | 0.0232 | 2.9840 | 0.0234 | 3.9853 | 93.80% | 0.5616 | 0.5655 | 0.5752 |
| | | | | $\alpha_3$ | 9.0436 | 8.0346 | 8.0055 | 0.0240 | 8.0025 | 0.0240 | 7.9965 | 0.0239 | 12.3747 | 95.00% | 0.5922 | 0.5976 | 0.6043 |
| | | | | $\lambda$ | 2.0710 | 0.1091 | 2.0011 | 0.0115 | 1.9991 | 0.0114 | 1.9952 | 0.0114 | 1.2656 | 93.80% | 0.4100 | 0.4117 | 0.4110 |
| | | | | R | 0.5497 | 0.0039 | 0.5554 | 0.0006 | 0.5558 | 0.0006 | 0.5566 | 0.0006 | 0.2457 | 95.00% | 0.0902 | 0.0905 | 0.0910 |
| | | 17, 22, 14 | 15, 20, 12 | $\alpha_1$ | 0.7570 | 0.0405 | 0.6884 | 0.0093 | 0.6870 | 0.0093 | 0.6842 | 0.0094 | 0.7854 | 97.00% | 0.3788 | 0.3766 | 0.3752 |
| | | | | $\alpha_2$ | 3.3983 | 1.1182 | 3.0062 | 0.0146 | 3.0044 | 0.0146 | 3.0007 | 0.0145 | 3.8439 | 95.00% | 0.4559 | 0.4555 | 0.4568 |
| | | | | $\alpha_3$ | 9.0610 | 5.5269 | 8.0105 | 0.0151 | 8.0088 | 0.0150 | 8.0054 | 0.0149 | 12.9750 | 97.00% | 0.4761 | 0.4733 | 0.4721 |
| | | | | $\lambda$ | 2.1042 | 0.0983 | 2.0007 | 0.0094 | 1.9993 | 0.0094 | 1.9966 | 0.0094 | 1.1605 | 95.00% | 0.3820 | 0.3808 | 0.3814 |
| | | | | R | 0.5588 | 0.0039 | 0.5572 | 0.0004 | 0.5575 | 0.0004 | 0.5580 | 0.0004 | 0.2452 | 97.00% | 0.0759 | 0.0757 | 0.0755 |
| | 30, 40, 30 | 25, 30, 22 | 20, 24, 20 | $\alpha_1$ | 0.7416 | 0.0243 | 0.7076 | 0.0110 | 0.7057 | 0.0109 | 0.7019 | 0.0107 | 0.5900 | 95.40% | 0.4090 | 0.4082 | 0.4065 |
| | | | | $\alpha_2$ | 3.2292 | 0.7106 | 2.9957 | 0.0202 | 2.9931 | 0.0202 | 2.9878 | 0.0203 | 3.1832 | 94.80% | 0.5459 | 0.5466 | 0.5530 |
| | | | | $\alpha_3$ | 8.7951 | 6.3867 | 8.0028 | 0.0251 | 7.9997 | 0.0251 | 7.9935 | 0.0251 | 10.1981 | 95.40% | 0.5956 | 0.6037 | 0.6049 |
| | | | | $\lambda$ | 2.0479 | 0.0693 | 1.9991 | 0.0109 | 1.9972 | 0.0109 | 1.9936 | 0.0109 | 1.0160 | 94.80% | 0.4072 | 0.4057 | 0.4051 |
| | | | | R | 0.5510 | 0.0024 | 0.5534 | 0.0004 | 0.5537 | 0.0004 | 0.5544 | 0.0004 | 0.1919 | 95.40% | 0.0787 | 0.0792 | 0.0793 |
| | | 27, 35, 26 | 23, 30, 23 | $\alpha_1$ | 0.7443 | 0.0228 | 0.6970 | 0.0087 | 0.6956 | 0.0087 | 0.6928 | 0.0087 | 0.5634 | 94.80% | 0.3635 | 0.3610 | 0.3605 |
| | | | | $\alpha_2$ | 3.2176 | 0.6404 | 2.9913 | 0.0140 | 2.9895 | 0.0141 | 2.9860 | 0.0143 | 3.0219 | 96.20% | 0.4573 | 0.4604 | 0.4599 |
| | | | | $\alpha_3$ | 7.9332 | 4.1515 | 7.9981 | 0.0144 | 7.9964 | 0.0144 | 7.9931 | 0.0144 | 10.5878 | 94.80% | 0.4555 | 0.4552 | 0.4541 |
| | | | | $\lambda$ | 2.0655 | 0.0674 | 2.0035 | 0.0079 | 2.0023 | 0.0079 | 1.9998 | 0.0079 | 0.9862 | 96.20% | 0.3512 | 0.3504 | 0.3508 |
| | | | | R | 0.5533 | 0.0023 | 0.5554 | 0.0003 | 0.5557 | 0.0003 | 0.5562 | 0.0004 | 0.1895 | 94.80% | 0.0706 | 0.0705 | 0.0707 |

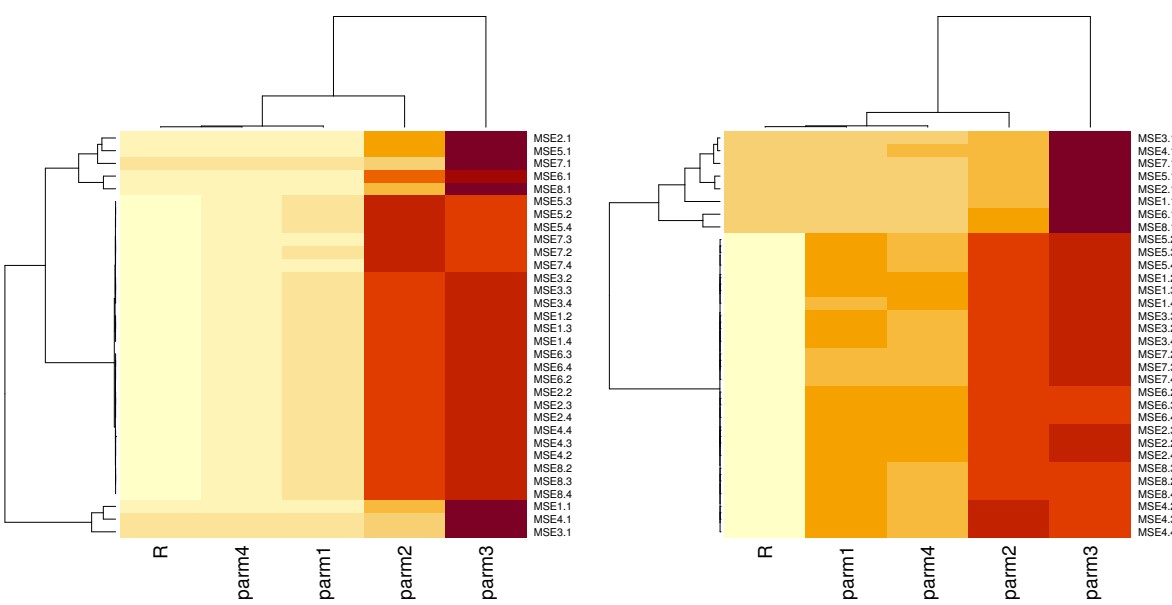

**Figure 3.** Heatmaps of MSE for parameters and R based on different estimation methods: Case III and IV.

## 6. Application of Transformer Insulation Data

This section examines an actual dataset in order to demonstrate the methodologies proposed in the previous sections. This dataset is also used to display the EED based on the GPHC method. The findings of a stress–strength life test of transformer insulation were published in chapter three of Nelson's book [45]. The test included three levels of voltage, which were 35:4 kv, 42:4 kv, and 46:7 kv, respectively, with a normal voltage of 14:4 kv. When 42:4 kv, the data are 0.6, 13.4, 15.2, 19.9, 25.0, 30.2, 32.8, 44.4, and 56.2. When 46:7 kv, the data are 3.1, 8.3, 8.9, 9.0, 13.6, 14.9, 16.1, 16.9, 21.3, and 48.1. When 35:4 kv, the data are 40.1, 59.4, 71.2, 166.5, 204.7, 229.7, 308.3, and 537.9. We want to calculate the reliability of the stress–strength model $P(X < Y < Z)$. Firstly, Table 6 provides the values of the estimate with different measures as the Kolmogorov–Smirnov (KS) statistic along with its *p*-value, Akaike information criterion (AIC), Bayesian information criterion (BIC), corrected AIC (CAIC), and Hannan–Quinn information criterion (HQIC) for the EED based on each sample of the dataset. The EED fits each dataset according to the KS test and the accompanying *p*-value. We have also included two plots based on the estimated model. On the left, the estimated and the empirical CDF of the EED are displayed, and on the right, the P-P plot estimated of the EED is displayed, see Figures 4–6 for each dataset, respectively. Figures 4–6 prove the fitting of each sample to show how each sample fits into the EED. The ML and Bayesian estimators via the complete sample are listed in Table 7. We notice that the standard error (SE) of the Bayesian estimates is lower than others for the ML estimates. Thus, the Bayesian estimation is the optimum estimation of the parameters for the EED based on the GPHC. The reliability of $P(X < Y < Z)$ of the Bayesian estimation method is higher than that of the ML process, proving the stated conclusion.

**Table 6.** ML and Bayesian estimates for reliability in stress–strength model based on complete sample.

|  |  | Estimates | SE | KS | *p*-Value | AIC | CAIC | BIC | HQIC |
|---|---|---|---|---|---|---|---|---|---|
| X | $\alpha_1$ | 1.3044 | 0.5729 | 0.2396 | 0.5994 | 80.5804 | 82.5804 | 80.9749 | 79.7292 |
|  | $\lambda$ | 0.0443 | 0.0173 |  |  |  |  |  |  |
| Y | $\alpha_2$ | 4.7666 | 2.7156 | 0.1927 | 0.8322 | 60.9491 | 62.9491 | 61.3435 | 60.0979 |
|  | $\lambda$ | 0.1769 | 0.0537 |  |  |  |  |  |  |
| Z | $\alpha_3$ | 1.8149 | 0.9348 | 0.1901 | 0.8864 | 103.7330 | 106.1330 | 103.8919 | 102.6614 |
|  | $\lambda$ | 0.0070 | 0.0028 |  |  |  |  |  |  |

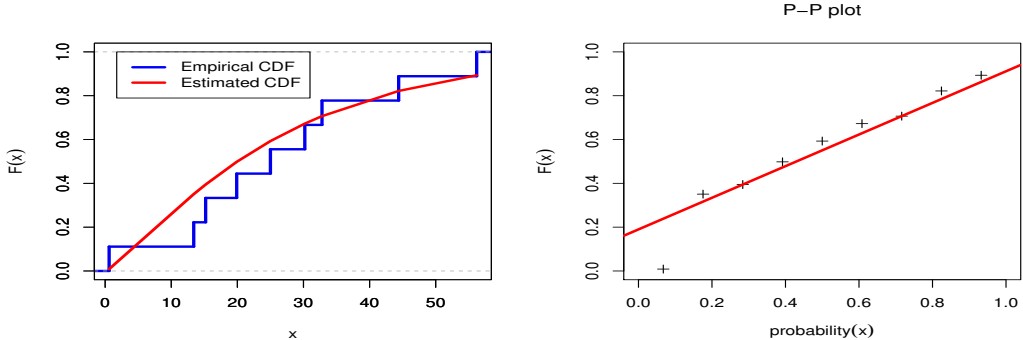

**Figure 4.** Empirical CDF and P-P plots for the EED for dataset 1.

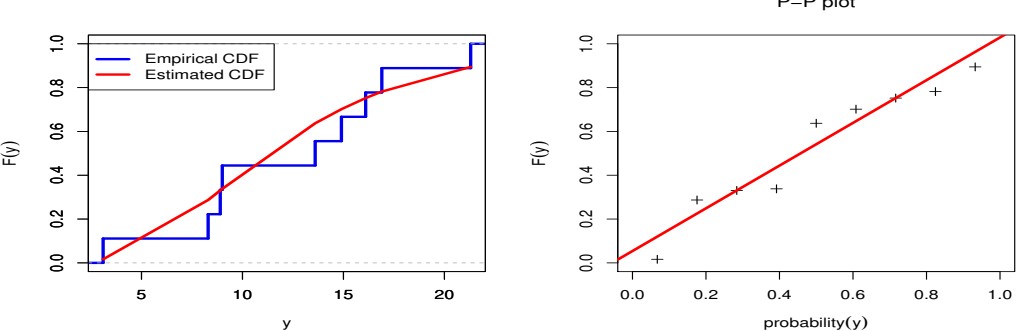

**Figure 5.** Empirical CDF and P-P plots for the EED for dataset 2.

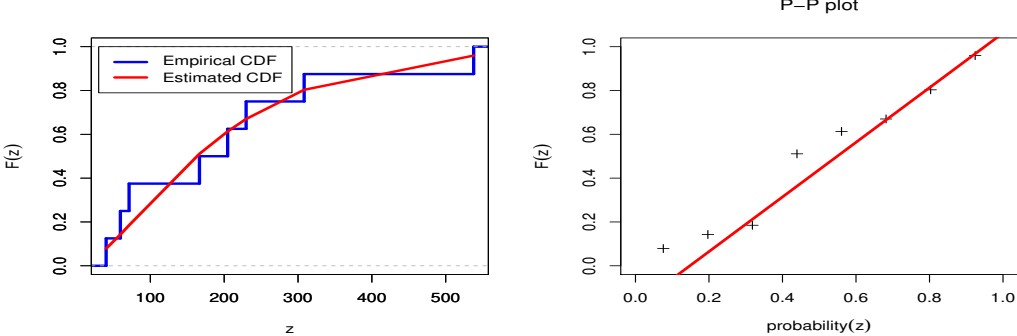

**Figure 6.** Empirical CDF and P-P plots for the EED for dataset 3.

It is first assumed that $X \sim EED(\lambda, \alpha_1)$, $Y \sim EED(\lambda, \alpha_2)$, and $Z \sim EED(\lambda, \alpha_3)$. Following that, these are the ML estimates for the unknown parameters: $\alpha_1 = 0.6089$, $\alpha_2 = 0.4964$, $\alpha_3 = 3.5237$, and $\lambda = 0.0131$; and $L_0 = -129.9995$ for the corresponding log-likelihood value. Second, assume that $X \sim EED(\lambda_1, \alpha_1)$, $Y \sim EED(\lambda_2, \alpha_2)$, and $Z \sim EED(\lambda_3, \alpha_3)$. The following are the ML estimates for the unknown parameters: $\alpha_1 = 1.3049$, $\alpha_2 = 4.7632$, $\alpha_3 = 1.8156$, $\lambda_1 = 0.0443$, $\lambda_2 = 0.1768$, and $\lambda_3 = 0.0071$; and $L_1 = -116.6313$ for the corresponding log-likelihood value. The following tests of the hypotheses are conducted by us:

$$H_0 : \lambda_1 = \lambda_2 = \lambda_3; \quad H_1 : \lambda_1 \neq \lambda_2 \neq \lambda_3$$

Then, the likelihood ratio tests value is $-2(L_0 - L_1) = -26.73644$ in this instance, with degrees of freedom as 2 (difference in the number of parameters for the two models). Moreover, the *p*-value of the Chi-squared test is one; therefore, the null hypothesis cannot be rejected. As a result, $H_0$ is a valid assumption in this situation.

**Table 7.** ML and Bayesian estimates for stress–strength model based on complete sample.

| | ML | | | | Bayesian | | | |
|---|---|---|---|---|---|---|---|---|
| | Estimates | SELF | Lower | Upper | Estimates | SELF | Lower | Upper |
| $\alpha_1$ | 0.6170 | 0.2216 | 0.1827 | 1.0513 | 0.6506 | 0.1647 | 0.2896 | 1.0329 |
| $\alpha_2$ | 0.5031 | 0.1781 | 0.1540 | 0.8523 | 0.5416 | 0.1716 | 0.2189 | 0.8249 |
| $\alpha_3$ | 3.6845 | 1.7466 | 0.2612 | 7.1077 | 4.0167 | 1.6591 | 1.0943 | 7.0266 |
| $\lambda$ | 0.0136 | 0.0035 | 0.0067 | 0.0204 | 0.0145 | 0.0033 | 0.0082 | 0.0202 |
| R | 0.3445 | | | | 0.3503 | | | |

Figure 7 depicts the trace and normal curve of the posterior distribution for the MCMC estimate of the stress–strength model for the EED based on the GPHC. The MCMC samples are shown as a pairs plot in Figure 8, which displays the pairwise relationship between the parameters in the top plot, the correlation coefficients in the bottom plot, and the marginal frequency distribution for each parameter on the diagonal. Moreover, as shown in Figure 9, for the reliability stress–strength estimate for the EED based on the complete sample, convergence begins at 2000 iterations or fewer. The MCMC samples are displayed as a pairs plot in Figure 8, which depicts the pairwise relationship between the parameters as independent, with the scatter plots matrix in the top plot, the correlation coefficients in the bottom plot, and the marginal frequency distribution for each parameter on the diagonal. The parameters P3 and P4 are shown to be medially connected in this diagram, where p1 is an $\alpha_1$, p2 is an $\alpha_2$, p3 is an $\alpha_3$, and p4 is a $\lambda$.

For each component of this model, we suggested using the following GPHC sample: $X$ = (0.6, 13.4, 15.2, 19.9, 25.0, 30.2, 32.8, 44.4), $Y$ = (3.1, 8.3, 8.9, 9.0, 13.6, 14.9, 16.1), $Z$ = (40.1, 59.4, 71.2, 166.5, 204.7, 229.7, 308.3), $R_1$ = (0, 0, 0, 0, 0, 0, 0, 1, 0), $R_2$ = (0, 0, 0, 0, 0, 0, 0, 0, 1), and $R_3$ = (0, 0, 0, 0, 0, 0, 0, 2).

The ML and Bayesian estimations of the EED parameters based on the GPHC sample of the stress–strength model are discussed in Table 8. The trace and normal curve of the posterior distribution for the MCMC estimate of the stress–strength for the EED based on the GPHC are shown in Figure 10. The MCMC samples are given as a pairs plot in Figure 11, which illustrates the pairwise relationship between the parameters in the top plot, the correlation coefficients in the bottom plot, and the marginal frequency distribution for each parameter on the diagonal. Moreover, as shown in Figure 12, for the reliability stress–strength estimate of the stress–strength model for the EED based on the censored sample, convergence begins at 2000 iterations or less.

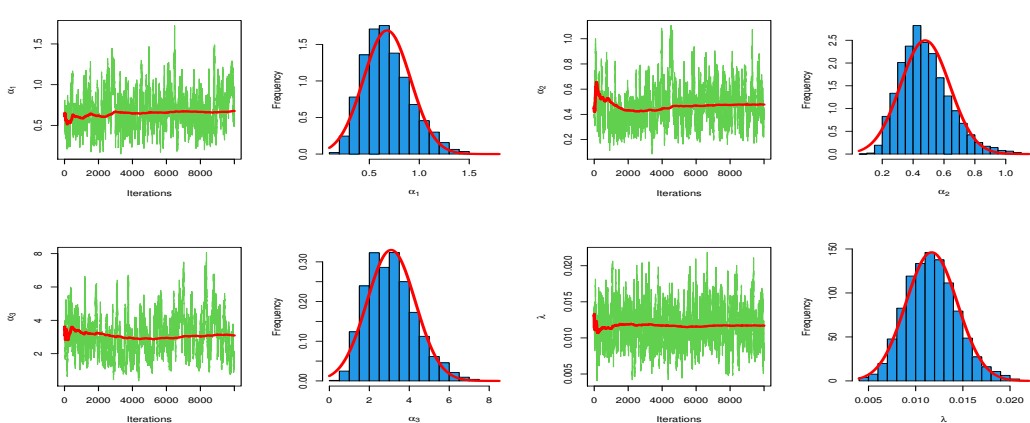

**Figure 7.** Trace and normal curve of posterior distribution for MCMC estimation of stress–strength for EED based on complete sample.

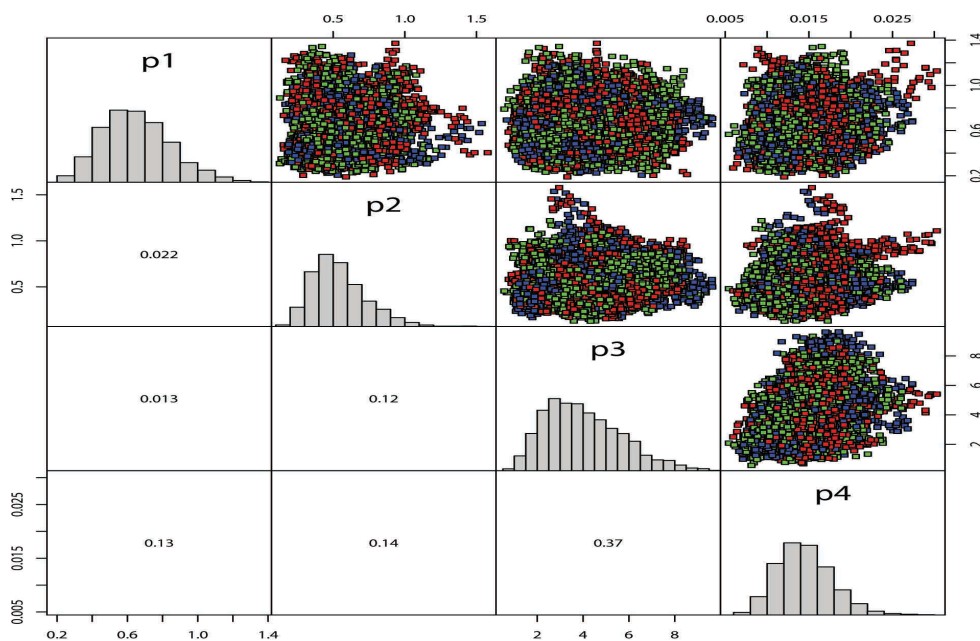

**Figure 8.** Pairs plots of the MCMC samples for parameter estimates of stress–strength for EED based on complete sample.

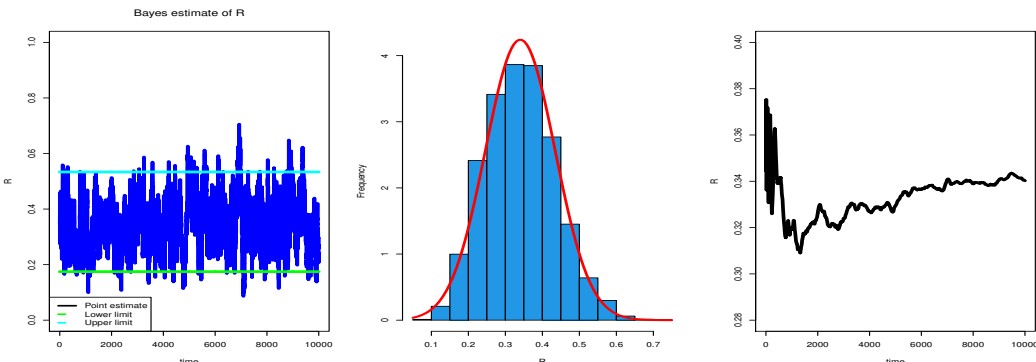

**Figure 9.** Trace and normal curve of posterior distribution for MCMC estimation of reliability stress–strength for EED based on complete sample.

**Table 8.** ML and Bayesian estimates for reliability of stress–strength model.

| $T_1, T_2, T_3$ | | ML | | | | Bayesian | | | |
|---|---|---|---|---|---|---|---|---|---|
| | | Estimates | SELF | Lower | Upper | Estimates | SELF | Lower | Upper |
| | $\alpha_1$ | 0.5577 | 0.2012 | 0.1633 | 0.9520 | 0.5817 | 0.1833 | 0.2593 | 0.9484 |
| | $\alpha_2$ | 0.5039 | 0.1797 | 0.1517 | 0.8562 | 0.5618 | 0.1739 | 0.2327 | 0.8495 |
| 25,60,550 | $\alpha_3$ | 2.7259 | 1.3152 | 0.1481 | 5.3037 | 3.4225 | 1.5595 | 0.9334 | 5.6519 |
| | $\lambda$ | 0.0105 | 0.0031 | 0.0044 | 0.0166 | 0.0118 | 0.0030 | 0.0062 | 0.0168 |
| | R | 0.3416 | | | | 0.3683 | | | |
| | $\alpha_1$ | 0.5726 | 0.2067 | 0.1674 | 0.9778 | 0.6069 | 0.2007 | 0.2482 | 0.9376 |
| | $\alpha_2$ | 0.5116 | 0.1826 | 0.1537 | 0.8695 | 0.5740 | 0.1720 | 0.2101 | 0.8999 |
| 20,50,500 | $\alpha_3$ | 2.9021 | 1.4057 | 0.1468 | 5.6573 | 3.6067 | 1.3633 | 1.0457 | 5.1271 |
| | $\lambda$ | 0.0111 | 0.0032 | 0.0047 | 0.0174 | 0.0124 | 0.0023 | 0.0067 | 0.0169 |
| | R | 0.3435 | | | | 0.3662 | | | |

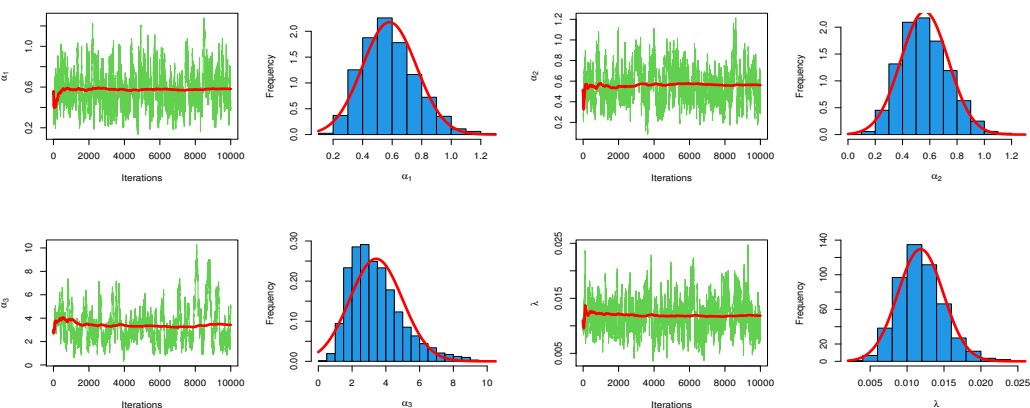

**Figure 10.** Convergence plots of MCMC for parameter estimates of the EED for censored sample.

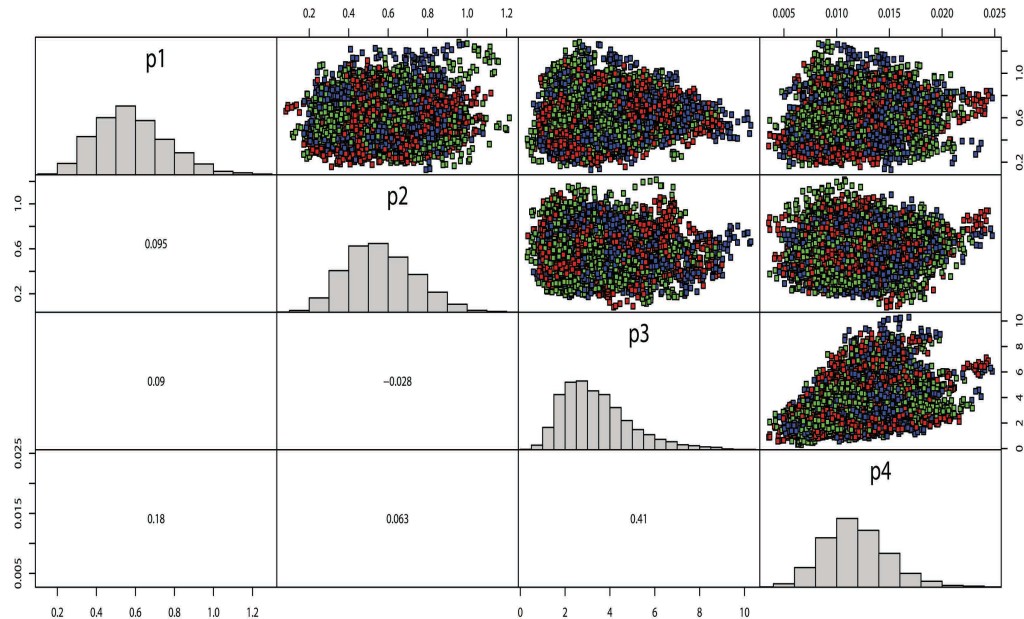

**Figure 11.** Pairs plot of the MCMC samples for parameter estimates of stress–strength for EED based on complete sample.

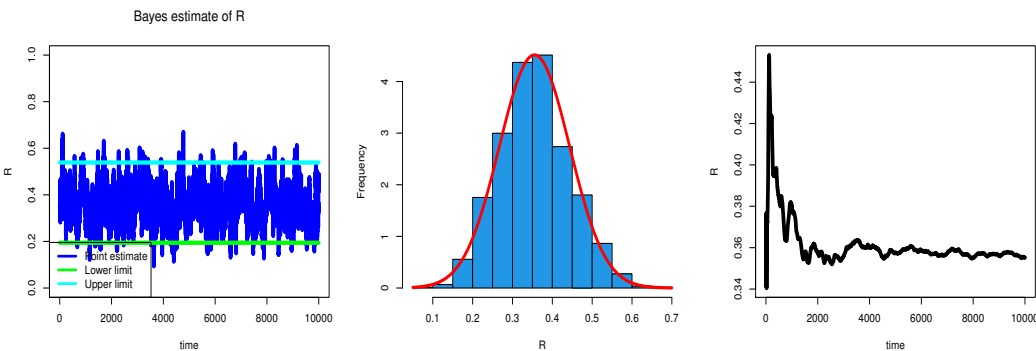

**Figure 12.** Trace and normal curve of posterior distribution for MCMC estimation of reliability stress–strength for EED based on censored sample.

## 7. Summary and Conclusions

In this study, the statistical inference of $R = P(X < Y < Z)$ for a component which has a strength that is independent on opposite lower and upper bound stresses, when the stresses and strength follow EED is discussed. We assume that both stresses and strength random variables are independent, having EED with a common scale parameter. Using the generalized progressive hybrid censoring design, various point and interval estimators for the reliability model $R$ are obtained in view of the ML and Bayesian approaches. Bayesian estimators are achieved by the MCMC method, as well as the MH algorithm, based on SELF and Linex loss functions, which are all conducted in light of informative priors. The CIs are derived using asymptotic distribution theory as well as Bayes credible intervals being constructed. The Monte Carlo simulation is conducted for the comparison of the effectiveness of the proposed estimates through some measures, such as average values, MSE, and CIs lengths. The outcomes of the study showed that the Bayes estimates produce a lower MSE for four parameters and stress–strength reliability based on a generalized progressive hybrid censoring scheme. Finally, a progressively censored real dataset is presented for illustration purposes. More in-depth studies, such as the application of the Bayesian estimation for the exponentiated exponential distribution based on various schemes, should be discussed in the future.

**Author Contributions:** Conceptualization, M.M.Y., H.M.A. and A.-A.H.E.-B.; methodology, M.M.Y., A.S.H. and E.M.A.; software, M.M.Y., A.S.H. and E.M.A.; validation, A.S.H., H.M.A. and A.-A.H.E.-B.; formal analysis, M.M.Y., A.S.H., E.M.A. and H.M.A.; resources, A.S.H. and E.M.A.; data curation, A.S.H., H.M.A., A.-A.H.E.-B. and E.M.A.; writing—original draft preparation, M.M.Y., E.M.A. and A.S.H.; writing—review and editing, M.M.Y., E.M.A., A.S.H., H.M.A. and A.-A.H.E.-B.; funding acquisition, H.M.A. All authors have read and agreed to the published version of the manuscript.

**Funding:** Princess Nourah bint Abdulrahman University Researchers Supporting Project number (PNURSP2022R299), Princess Nourah bint Abdulrahman University, Riyadh, Saudi Arabia.

**Institutional Review Board Statement:** Not applicable.

**Informed Consent Statement:** Informed consent was obtained from all subjects involved in the study.

**Data Availability Statement:** Datasets are available in the application section.

**Acknowledgments:** The authors thank the support from the Princess Nourah bint Abdulrahman University Researchers Supporting Project number (PNURSP2022R299), Princess Nourah bint Abdulrahman University, Riyadh, Saudi Arabia. We thank the referees for their valuable suggestions which improved the paper.

**Conflicts of Interest:** The authors declare no conflict of interest.

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
