# Peer review of "Bayesian and Non-Bayesian Analysis of Exponentiated Exponential Stress–Strength Model Based on Generalized Progressive Hybrid Censoring Process"

_axioms, doi:10.3390/axioms11090455_

Round 1
Reviewer 1 Report
See attachment.

Author Response
First of all, we would like to express my sincere thanks and appreciation to the reviewers for their comments, which improved the paper. All the corrected parts are written with red color within the main text.
Comments:
(1). Comparing with common competing risks model R = P(X < Y), the background and motivation of this paper about the R = P(X < Y < Z) should be given more explanation, even about its practical applications.
Answer
More application of the model added with red color as background
In application section more explanations are added with red color
===============================
(2) It is better to rewritten the contents of the Section One for a contextual expression, where the origin contents are too long and written loosely.
Answer
According to comment 2 and 3, section one is shortened
===============================
(3) The description of the progressive type-II censoring is not necessary.
Answer
Ok, we delete it
===============================
(4) For giving a tight expression, I suggest to remove the GPHC scheme to Section two, and the name is changed as “data description and reliability model” for clearness.
Answer
Ok thanks, we done it
===============================
(5) In this paper, it is assumed that stress and strength variables X, Y and Z follow EED models with common scale parameter l. However, such assumption sometimes does not always available from practical perspective. Please explain why take common scale parameters in this analysis, and how about the case with unequal li, i= 1,2,3?.
Answer
We explain the answer for this comment in section 6 as follows: .
It is first assumed that X ∼ EED(λ, α1), Y ∼ EED(λ, α2), and Z ∼ EED(λ, α3). Following that, these are the ML estimates for the unknown parameters: α1 =0.6089, α2 =0.4964, α3 =3.5237, and λ =0.0131, and L0 = -129.9995 for the corresponding log-likelihood value.
Second, assumed that X ~ EED( ), Y ~ EED( , α2), and Z ∼ EED(λ3, α3). The following are the ML estimators for the unknown parameters: α1 =1.3049, α2 =4.7632, α3 =1.8156, λ1 =0.0443, λ2 =0.1768, and λ3 =0.0071; and L1 =-116.6313 for the corresponding log-likelihood value.
The following tests of hypotheses are conducted by us:
H0: λ1 = λ2 = λ3; H1: λ1 λ2 λ3
Then the likelihood ratio tests value is −2(L0 − L1) =-26.73644 in this instance with degrees of freedom is 2 (difference in the number of parameters for the two models). Also the
P-value of chi-squared test is 1, therefore this; the null hypothesis cannot be rejected. As a
result, H0 is a valid assumption in this situation
===============================.
(6) For solving MLEs, the authors proposed Newton’s iteration method in this paper without any commenting on? how complex is it to do this numerically? What about complex for other methods? The estimation procedures presented in the manuscript are standard parameter estimation methods. Is there any convergence problem in estimation procedure for the proposed methods? and how fast does the algorithm converge to targeted estimates in computational approach? Details about methods, packages, computational efforts etc. need to be given. Authors should provide more technique details.
Answer
According to above comments we add the following:
In subsection 3.1 we add the following
There are no analytical solutions for the unknown parameters ,and from Equations 8. As a result, Equations 8 may be maximised using a fairly straightforward iterative approach like the Newton-Raphson (NR) procedure to get the appropriate MLEs of ,and . The corresponding precise distribution (or exact confidence intervals) of ,and cannot be obtained since the MLEs of ,and cannot be obtained in closed form. In order to do this, we advise using the "maxLik" package for the given dataset ,and and utilising the NR iterative approach for ML computations. Since the MLEs of ,and cannot be obtained in closed form, so the corresponding exact distribution (or exact confidence intervals) of ,and is not possible to obtain. For this purpose, we suggest to implement ‘maxLik’ package for any given dataset ,and , by using the N-R iterative method for ML calculations.
In simulation study, we add the following;
In the left side of Tables 1-4, the average of frequentist estimates and the average of Bayesian MCMC estimates of ,and are determined. Additionally, in the right side of Tables 1 to 4 report the ACLs of ,and that were obtained. Two highly regarded R software packages are used to conduct out extensive evaluations: the 'CODA' package for computing Bayes estimates using MCMC techniques and the 'maxLik' tool for computing MLE estimates using the NR algorithm.
=================================
(7) Please indicate why choose gamma priors in Bayesian analysis? is there something attractive reason for such assumption.
Answer
Gamma distribution, upon its parameter values, has many types of shapes, so gamma family of distributions is very flexible and can be considered as a suitable prior for any unknown parameter(s), see references Kundu (2008) and Dey et al. (2016).
==========================================
(8). From the Eq. (12), “the marginal posterior densities of the parameters , , and should conditional posterior densities, so the associated estimates should be obtained via Gibbs sampling along with Metropolis-Hastings technique ? So, please revise the iteration approach in the paper.
Answer
The marginal posterior densities in (12) are not well-known distributions, since the forms α1, α2, α3, and λ don’t take any known distributions, so we used the Metropolis–Hastings sampler to generate the values of α1, α2, α3, and λ with normal proposal distribution to generate samples from it in (12). Furthermore, the marginal posterior density of R is not well-known distribution, again we used the Metropolis–Hastings sampler.
==========================================
(9). For the real data example, How to choose the values of the hyper-parameters in priors, it does not mentioned in the paper.
Answer
To find out how to elicit hyper-parameters of the independent joint prior, we can use the likelihood method's estimate and variance-covariance matrix. The mean and variance of gamma priors can be used to represent the derived hyper-parameters (As mentioned in simulation section).
==========================================
(10). There are grammatical errors/typos and expressions in the writing. Some are provided as follows.
P2 L60, in the whole contents, notations “Type-I, type-II, type-I, Type-II” should be written in consistent way.
P8 L139": “So” ! “Therefore”.
P10 L19: “m” ! “m”.
P15 L228-230: data Z, Y,X should be presented in alphabet order as X, Y,Z.
P17 L245: a blank should be added before word “Using”.
Answer
Ok, thanks all are corrected
Best regards

Reviewer 2 Report
Please, see the attachment.

Author Response
First of all, we would like to express my sincere thanks and appreciation to the reviewers for their comments, which improved the paper. All the corrected parts are written with red color within the main text
Main Comments
The “Abstract” should be simplified better highlighting the problem addressed in the study and how it is dealt with (according to related studies). Any formula should be described and clarified in the “Introduction”.
Answer
Ok, we are simplified and rewritten the abstract with red color
==============================
The Introduction should be simplified. Any formula in detail and algorithms should be just described and – then – clarified in the methodology section.
Answer
Ok, we done it
==============================
In ‘Section 2’, the results in Figure 1 should be better clarified for non-expert readers.
Answer
Clarification of figure 1 is added with red color
==============================
In ‘Sections 3’ and ‘4’, some improvements are in order.
The brackets in equations (7) and (8) should be put on the left.
Better explain the usefulness of the asymptotic confidence interval obtained (discussing the hyperparameter ).
Further discussions would be well-liked in ‘Section 4’ describing Bayesian methods.
The results achieved in Figures 2-10 should be better specified according to related (previous) studies.
Answer
Ok, we put the brackets on the left
Done in subsection 3.2
More discussion and description are added in section4
Done
Best regards
Reviewer 3 Report
The pape titled "Bayesian and Non-Bayesian Analysis of the Exponential Exponential Exponential Stress Force Model Based on the Generalized Progressive Hybrid Censorship Process" presents a comparison between the maximum likelihood estimate and the Bayesian estimate for a reliability model given by GPHC. In general, this article is interesting, however I still do not find the advantage that this model has compared to the numerous models already published and shown in the introduction of this article, so I think it would be necessary to provide more clearly the advantages that this model has. On the other hand, in the comparison that is made between the MLE estimator and the Bayesian estimators, it is shown that the latter provide greater precision, since they have lower MSE; however, the focus is not too much on this issue, an issue that I consider of great importance, since it is optimal to always consider the most precise estimators, and on this occasion they are provided by the Bayesian strategy. Another question I would like to know is whether they have tried to consider other a priori distributions for the parameters, or just the Gamma distribution. Finally, I would recommend that tables 1-4 be displayed using graphs, as observing the values shown has great difficulty for the reader. I think a graph showing the MSE values between the different estimators, as well as their confidence intervals, or a graph showing the AvE estimates would be easier to differentiate.
Author Response
First of all, we would like to express my sincere thanks and appreciation to the reviewers for their comments, which improved the paper. All the corrected parts are written with red color within the main text.
Finally, I would recommend that tables 1-4 be displayed using graphs, as observing the values shown has great difficulty for the reader. I think a graph showing the MSE values between the different estimators, as well as their confidence intervals, or a graph showing the AvE estimates would be easier to differentiate.
Answer
Done, we plot Heatmaps of MSE with explanation of the plots
Round 2
Reviewer 2 Report
The paper has been improved. It is an interesting work and an enjoyable reading.